# Assessing European cities with the 3-30-300 rule underscores the need for enhanced urban greening efforts

L. E. Bertassello [1] ✉, M. van der Velde [1], J. Maes [2], S. Liu[3], M. Brandt [3] & L. Feyen [1]

Urban green spaces are fundamental to sustainable city living, providing essential temperature regulation and social well-being. However, rapid urbanization often threatens these areas, exacerbating socio-economic disparities in access. This study evaluates adherence to the 3-30-300 rule - a guideline advocating for three trees visible from every home, 30% neighborhood canopy cover, and a park within 300 meters - across 862 European cities. Here we show that less than 15% of the studied population lives in full accordance with these criteria, while 21% reside in areas that do not meet any of the three benchmarks. Our analysis reveals strong vegetation inequalities, where higher levels of urban greenness are consistently associated with wealthier settlements. These findings indicate that most European cities currently fall short of providing equitable access to nature, underscoring an urgent need for a paradigm shift in urban planning to sustainably and equitably address the demands of growing populations.

Cities are pivotal in driving societal change[1]. Within this context, urban green spaces significantly enhance quality of life by mitigating environmental stressors such as high temperatures[2,3], noise pollution[4], and poor air quality[5], while simultaneously fostering health and social well-being[6,7]. Despite these critical benefits, ongoing urbanization frequently leads to the loss of these vital spaces, thereby exacerbating socio-economic disparities in access to urban greenery[8]. As the global urban population expands, prioritizing proximity to green spaces and ensuring sufficient tree coverage becomes increasingly imperative to meet UN Sustainable Development Goal 11 Target 7, which advocates for universal availability of inclusive and accessible green areas[9]. In the context of a rapidly urbanizing world, concepts like the biophilic cities[10] or smart sustainable cities[11] urge urban planners to facilitate daily interactions with nature, promoting environmental awareness and stewardship. These cities are envisioned to be sustainable and resilient, strengthening their social and physical capacity against future shocks such as climate change, natural disasters, and economic uncertainties.

The European Union, EU, has experienced considerable urbanization in recent decades, with approximately 73% of its population now living within urban areas, a figure that exceeds the global average[12,13]. Building upon the European Green Deal's (EC 2019/640)[14] commitment to a green transition, the EU is actively promoting urban greening through key initiatives like the Green City Accord (EC 2021)[15], a voluntary commitment by cities to improve environmental quality and integrate nature. The legally binding Nature Restoration Regulation (EC 2024/1991)[16] further mandates that EU cities and towns prevent the loss of green spaces and tree cover by 2030 and demonstrate a continuous increase thereafter. To track progress, the EU has also established a system for ecosystem accounting[17], enabling the measurement and monitoring of urban green space and the benefits they provide.

In parallel with these legislation and policy initiatives, Konijnendijk (2023)[18] launched a rule of thumb for urban forestry and urban greening: the 3-30-300 rule. This rule focuses on the crucial contributions of urban trees and other urban nature to health and well-

[1]European Commission, Joint Research Centre (JRC), Ispra, Italy. [2]European Commission, Directorate-General for Environment (DG-ENV), Brussels, Belgium. [3]Department of Geosciences and Natural Resource Management, University of Copenhagen, Copenhagen, Denmark. ✉e-mail: leonardo.bertassello@ec.europa.eu

being, as well as climate change adaptation. The multifaceted nature of the rule implicitly recognizes the need to consider many different aspects of city life to benefit from urban greenery. A key ambition is to let urban forests and other green spaces percolate into all of our living, working, and learning environments. Furthermore, the rule is easy to remember, and as we will show, relatively straightforward to implement, monitor, and evaluate. The 3-30-300 rule specifies that every dwelling, educational facility, and workplace should have a view of 3 trees, 30% canopy in the neighborhood, and a park within a 300-meter walk.

The 3-30-300 rule provides a set of objectives, universal benchmarks for cities to determine how they fare on urban green space visibility, coverage, and proximity, as well as where to target investments for the future. Since its introduction, the 3-30-300 rule has attracted considerable interest and numerous cities globally, including Malmo (Sweden), Zurich (Switzerland), St. Petersburg (Florida, USA), Haarlem (Netherlands), Saanich (Canada), and Viladecans (Spain), have incorporated the rule into their urban greening strategies, either formally or informally. Despite its quite rapid diffusion, the lack of a comprehensive assessment of the 3-30-300 rule's implementation presents a critical gap, with only a handful of studies evaluating the rule's fulfillment in cities[19–23]. Without robust evaluation, cities risk implementing strategies that fall short of effectively addressing disparities in green space access or achieving broader environmental goals. Urban densification strategies, in particular, have sometimes resulted in uneven distribution of greening efforts[24], with some instances contributing to green gentrification[8]. In these cases, well-intentioned greening initiatives inadvertently lead to the displacement of low-income and minority residents, highlighting the need for careful assessment and planning to ensure that urban greening benefits all community members equitably.

To address this gap, here, we evaluate the 3-30-300 rule across Europe, encompassing 862 cities with at least 50,000 inhabitants in the 27 EU countries, Iceland, Norway, Switzerland, and the UK (refer to Supplementary Table 1 for details). Our approach integrates innovative spatial analytics with up-to-date data on urban green spaces from the LUISA (Land Use-based Integrated Sustainability Assessment) Base Map[25], tree density from the Copernicus Land Monitoring Service[26], and individual tree counts from high-resolution satellite imagery[27,28] alongside population density and building height information from the Global Human Settlement Layer[29,30].

## Results and Discussion

### Nearly half of the residents in the studied cities meet the three-tree visibility criterion

Konijnendijk (2023)[18] suggests that every resident should have a view of three trees from their dwelling, educational facility, and workplace. However, defining this visibility is challenging. Various studies employ different methods: some use surveys and window-view analyses to assess tree visibility from households[31,32], others utilize computer vision to quantify street greenery[33], and some define tree visibility using buffer zones with distance thresholds[19,20,34]. Here, to assess the visibility of three trees, we used viewshed analysis[31,35,36] (see Methods). First, we mapped areas visible from specific observation points, accounting for obstructions like building height and surrounding terrain features. We then integrated these maps with high-resolution (3 m) tree count data from PlanetScope[27,28] (see Methods) to focus on raster cells containing at least three trees. Finally, we applied this layer of visual green exposure to the Global Human Settlement Layer Population[37] (GHS-POP) data to estimate the population residing in areas with visibility of at least three trees.

We find that about 46.7% ± 18% (standard deviation, SD) of the total urban population in the analyzed cities views at least 3 trees (Fig. 1a−d). In approximately 30 cities, including Lugano (Switzerland), Espoo (Finland), and Berlin (Germany), over 70% of residents view more than three trees. Conversely, in around 30 cities - such as Murcia (Spain), Ragusa (Italy), and Valletta (Malta) - fewer than 10% of the population meet this criterion. Our results reveal significant spatial variability in the adherence to this rule across European cities, the largest when compared to the other two criteria of the 3-30-300 guideline (see Supplementary Fig. 1). Southern European cities - particularly those in Spain, southern Italy, and Greece - generally demonstrate lower compliance with the 3 visible trees rule.

### One-quarter of the analyzed population lives in areas with 30% tree cover

To examine the second rule, we estimated the share of the urban population residing in neighborhoods with over 30% tree cover. Due to the lack of a standardized cross-city neighborhood definition, we adopted 1 km grid blocks to delineate neighborhoods (see Methods). This selection is consistent with prior literature that frequently utilizes a 500 m egocentric radius to define local environments[38–40]. Our analysis reveals that the 30% rule is, on average, the least fulfilled of the three rules, with about 28% of the population in the analyzed cities

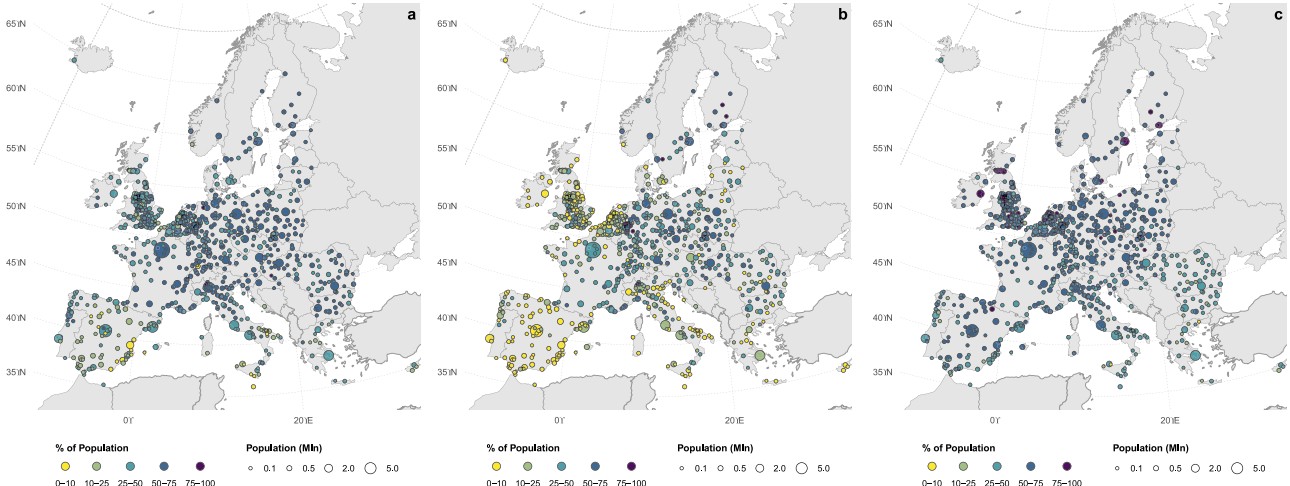

**Fig. 1 | Spatial distribution of 3-30-300 rule fulfillment across 862 European cities.** Maps (**a**), (**b**) and (**c**) illustrate the proportion of the urban population meeting the 3-tree, 30% canopy cover, and 300-meter proximity criteria, respectively. Circle dimensions are scaled to total city population, while the discrete color gradient indicates the percentage of residents satisfying each specific guideline.

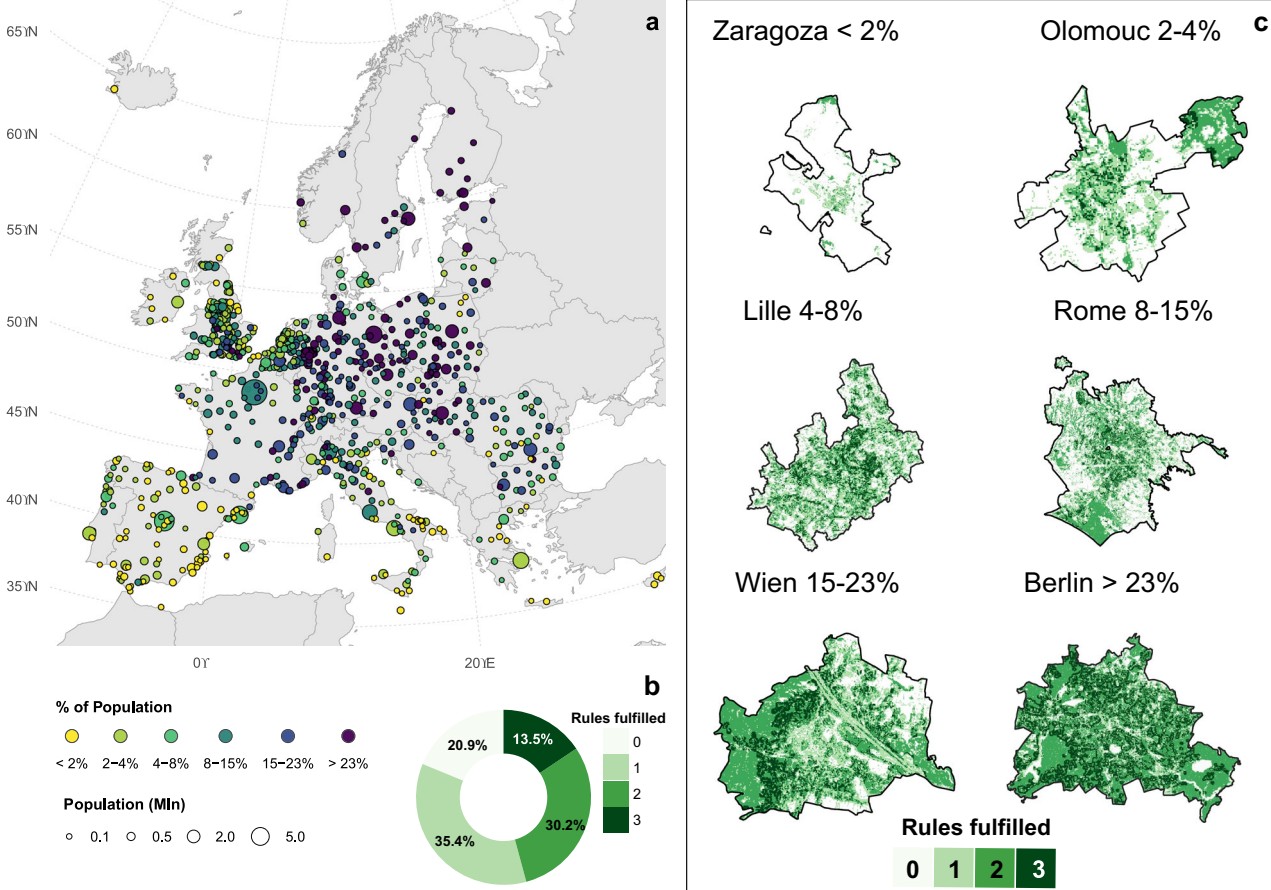

**Fig. 2 | Comprehensive assessment of 3-30-300 guideline fulfillment and intra-urban variability. a** Spatial distribution of guideline adherence across 862 European cities; circle size is proportional to total population, with color coding indicating the degree of fulfillment. **b** Summary of the aggregate population share

across all 862 cities meeting 0, 1, 2, or 3 criteria. **c** High-resolution analysis of six sample cities at a 100 m x 100 m grid level, illustrating intra-urban variability and identifying specific localized areas where multiple rules are simultaneously satisfied or missed.

residing in areas with at least 30% tree cover (Fig. 1b–e). At the same time, it is also the rule with the largest variability (SD = 25%), with cities located in northern and eastern Europe having a larger share of citizens living in green neighborhoods. For example, in cities like Berlin (Germany), Stockholm (Sweden) and Warsaw (Poland), more than 70% of the population resides in neighborhoods with high tree cover density. In stark contrast, one out of three European cities with more than 50,000 inhabitants has less than 10% of its population residing in green neighborhoods. The layout of green areas throughout the city strongly affects the outcome of this rule. For example, cities like Savona (Italy) and Baia Mare (Romania) have high tree cover density (85% for Savona and 82% for Baia Mare), yet only a small percentage of their populations live in compliance with the 30-rule (36% for Savona and 34% for Baia Mare). This highlights that a more even distribution of green spaces across a city leads to better compliance with urban green coverage.

### Over half of the residents in the analyzed cities live within 300 meters of a park

To examine the third rule, we utilized spatial buffering to estimate the proportion of the population residing within a 300-meter zone surrounding parks and other urban green areas larger than 1 ha (see Methods). This buffer serves as an indicator of the accessible green area for nearby residents. We find that 57% ± 14% (SD) of urban dwellers in the analyzed cities can access a green space (e.g., park) within 300 m distance (Fig. 1c–f), which makes this rule the most fulfilled of the three rules. Compliance with this rule also shows noticeable geographical

disparity. While more than 60% of European cities have over half of their population residing near green spaces, in northern cities like Stockholm (Sweden), Helsinki (Finland), Hastings (UK), and Zoetermeer (Netherlands), this exceeds 75%. Conversely, southern and south-eastern European cities possess smaller shares of green areas exceeding 1 ha and consequently display lower access to urban green spaces, resulting in less than 10% of the population having access to urban green areas within 300 m distance in cities such as Bagheria (Italy), Xanthi (Greece), and Arrecife (Spain).

### Less than 15% of the population in the analyzed cities fulfills the 3-30-300 rule

The overall fulfillment of the 3-30-300 rule is contingent upon the interaction among its individual criteria. A city might achieve high scores in each criterion independently, yet exhibit lower overall fulfillment if the spatial regions defined by these criteria do not intersect. Overall, we find that only a small fraction, 13.5%, of the dwellers in the analyzed European cities live according to the 3-30-300 rule.

There are only two cities where more than 50% of the population satisfies the rule (Espoo in Finland and Varese in Italy), and about 20 cities where this percentage is above 40% (see Fig. 2a). With few exceptions, the largest share of populations satisfying all the three criteria are located in Scandinavian regions along with Germany and Poland with cities like Berlin, Stockholm, and, Helsinki having respectively 40%, 41%, and 43% of residents fulfilling the 3-30-300 rule. However, in 177, less than 1% of the residents fulfilled the three rules.

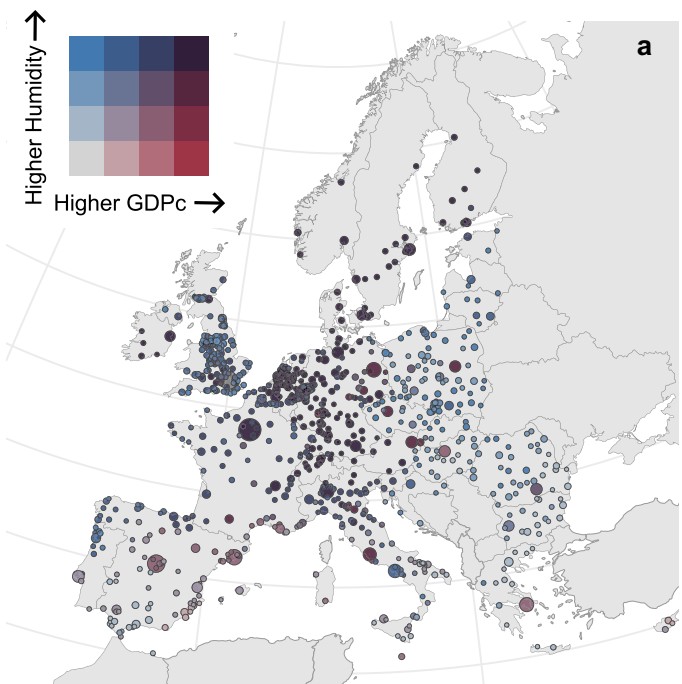

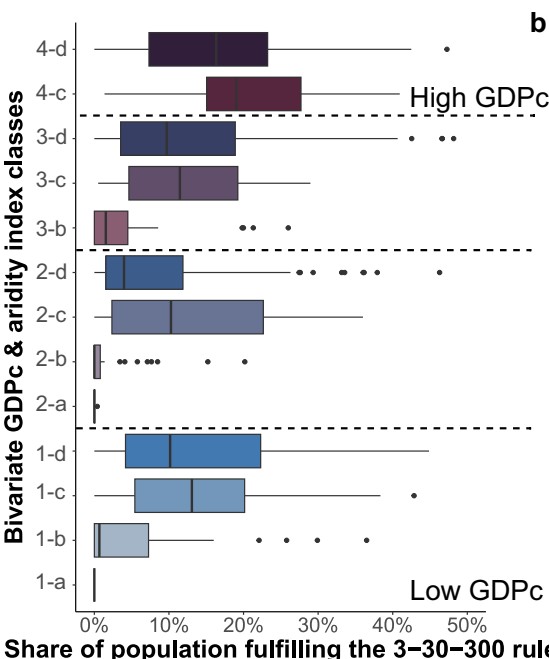

**Fig. 3 | Interplay between economic development, climate, and urban greenness across 862 European cities. a** Spatial bivariate distribution of per-capita GDP (GDPc) and aridity index; circle sizes are proportional to total city population, while color coding represents the 4th quantile (quadrants) of each variable's distribution. **b** Boxplot analysis of the 16 bivariate subgroups, correlating GDPc and aridity with the percentage of the population fulfilling the 3-30-300 rule. Horizontal dashed lines delineate the four GDPc quantiles, while the sub-groups illustrate the influence of increasing humidity on greenness. Note that each box represents the interquartile range (IQR) with a median line; whiskers extend to 1.5 x IQR, with individual points marking outliers. The data suggest a positive relationship between economic development and environmental quality, as higher-GDPc cities - often located in low-aridity regions - show a higher proportion of the population meeting the 3-30-300 criteria.

Our analysis reveals that areas within cities meeting all three criteria of the 3-30-300 rule are scarce. Approximately 21% of the analyzed urban population lives in areas that do not meet any of the 3-30-300 conditions. Furthermore, about 35% of the population resides in areas where only one condition is met, resulting in 56% of the total urban population of the 862 European cities living in locations where at most one condition is satisfied. These low percentages are primarily due to the limited proportion of the population living in neighborhoods with at least 30% tree cover, underscoring the difficulty for most cities in attaining the second rule. This finding is consistent with recent studies indicating that many cities fail to meet the 3-30-300 rule largely due to insufficient tree canopy[19–21].

There is significant spatial variability among cities concerning this metric (Fig. 2b). For example, in Berlin, the majority of the area (67%) meets at least two criteria of the 3-30-300 rule. In contrast, this proportion is lower in Vienna at 55%, resulting in only 20% of residents living in accordance with the 3-30-300 rule. Conversely, in Zaragoza, approximately 88% of the city's area does not meet any of the 3-30-300 rule's requirements, leading to less than 2% of the city's population satisfying the rule.

### Wealth influences access to urban greenery

The diverse benefits of urban green spaces are often undermined by their inequitable distribution. In this context, we find that there is a clear link between access to urban green and economic conditions (Fig. 3). Cities with higher per-capita GDP (GDPc) often provide more green space. This difference is especially visible in the spatial divide between central-northern European cities versus south-eastern cities. This trend is consistent with Chen et al., (2022)[41] who identified a contrasting difference of greenspace exposure between Global South and North cities, where wealthier cities experience about three times the greenspace exposure level than poorer cities. Another crucial factor influencing this relationship is humidity-aridity conditions. Cities located in more humid environments tend to have a greater share of green spaces, which results in a higher adherence to the 3-30-300 rule. Notably, we showed that on average, within the same GDP per capita classes, cities with higher humidity levels provide residents with more access to green spaces (Fig. 3b). It could be speculated that the greater access to green in cities in central-north Europe compared to southern-east cities might relate to more humid climatic conditions that favor the growth and proliferation of plants and urban green.

One potential limitation of using GDPc as a proxy for wealth is that it may fail to capture the real living standard of households[42]. Another limitation is that relying on a single metric to represent an entire city, such as Madrid, Paris, Rome, or Athens, masks smaller-scale heterogeneity. To address these limitations, we conducted additional analyses using high-resolution data on disposable income. First, we compared the proportion of the population meeting the 3-30-300 rule with a recently developed downscaled income dataset available at 1 × 1 km resolution, generated through an innovative machine learning framework[43]. Our results reveal a positive correlation between disposable income and the presence of green, suggesting that across Europe city-dwellers with higher disposable income tend to reside in urban areas with greater access to green spaces (Supplementary Fig. 2). Then, we performed a more granular analysis for 186 cities across France, Belgium, and Spain, using newly available high-resolution datasets on disposable income specific to these countries. The methodology is illustrated in Fig. 4a for the city of Lyon (France)

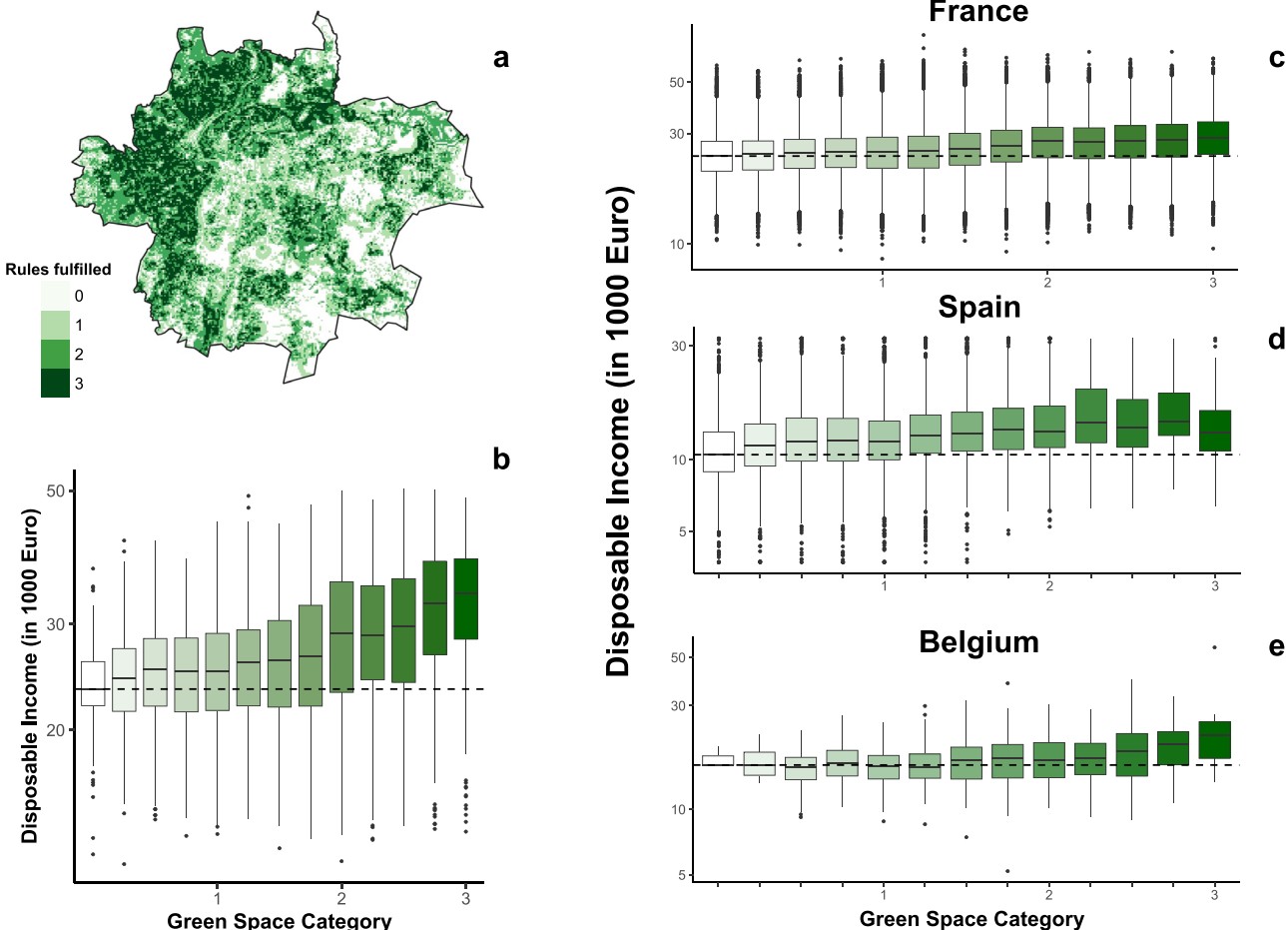

**Fig. 4 | Socioeconomic disparities in urban greenness access across France, Spain, and Belgium. a** High-resolution overlay of the FiLoSoFi (INSEE) 200 m x 200 m disposable income grid with the 3-30-300 rule compliance pixels for Lyon. **b** Correlation between the Compliance Score (Green Space Category) and the logarithm of disposable income for Lyon. The x-axis represents the average compliance level, calculated by summing the satisfied rules (0–3) for every pixel within a 200 m income grid cell and averaging the result. **c–e** Aggregated relationship between income and greenness for all analyzed cities in Belgium (*n* = 15), France (*n* = 72), and Spain (*n* = 99). Note that each boxplot illustrates the interquartile range (IQR) of income for each compliance category (0 = no rules met; 3 = all rules met). Medians are marked by a central line, and whiskers extend to 1.5 x IQR. The logarithmic scale on the y-axis highlights relative income disparities across the gradient of guideline fulfillment.

where disposable income data at 200 ×200 m were obtained from the National Institute of Statistics and Economic Studies, INSEE[44]. Examining data at such a finer scale not only corroborates broader trends but also enhances our understanding at the local level, offering a more nuanced and reliable insight into the relationship between income and urban greenness. Granular data significantly reduces aggregation bias, thereby providing an enhanced representation of the socioeconomic landscape, which is crucial for evaluating the impact of urban planning and development policies. The case of Lyon serves as a compelling example of this approach, illustrating the value of detailed analysis in capturing the complexities of urban dynamics. When assessed at the national level (Fig. 4b), a clear trend emerges across all three countries, reinforcing the consistent association between higher disposable income and the likelihood of residing in greener urban neighborhoods.

## Opportunities for urban green space development

The dynamic interplay between green infrastructure and urbanization presents a complex set of trade-offs that require careful consideration[45]. While urbanization is often seen as a positive force driving economic growth and resource efficiency, it can impact the availability and quality of urban green spaces if not managed thoughtfully. Addressing this challenge through intentional design and planning is crucial to creating sustainable and livable cities. The recent decade (2010-2020) has indeed witnessed a significant increase in urban population (+16% on average) and urban area expansion (+2.3% on average) within city boundaries in Europe (Supplementary Fig. 3). This urban growth has not been accompanied by a commensurate increase in green urban areas and tree cover density (Supplementary Fig. 3), with both indicators exhibiting stable or declining trends (UGA: −0.3% and TCD: −1.6%). This trend may be a contributing factor to the low percentage of the population in European cities living in satisfaction with the 3-30-300 rule. Without proper intervention, we risk further diminishing the already limited access to urban green areas.

In the current city configuration, nearly 90% of the population in the studied cities resides within 1 km of a park (Supplementary Fig. 4). By strategically integrating green spaces and parks within city planning, the distance to a park can be lowered. A key solution to achieving this balance is the transformation of urban mobility[46], a solution also advocated by the recent concept of the 15-minute city[47,48], which contends that cities can function more effectively, equitably, and environmentally if essential services and key amenities are within 15 minutes by non-polluting forms of transport, such as walking or cycling. Prioritizing such sustainable transport modes can significantly cut reliance on private vehicles, thereby lowering greenhouse gas emissions and freeing up valuable urban space typically used for roads and parking lots. This reclaimed space can be repurposed for green

areas, parks, and additional housing, fostering a more efficient and environmentally friendly cityscape. Importantly, there is a significant correlation ($p < 0.001$, rho = 0.39; see Supplementary Fig. 5) between adherence to the 3-30-300 rule and the opportunity to walk[49]. This correlation underscores the benefits of nature exposure in reducing stress and enhancing well-being, as walking in natural settings significantly decreases self-reported stress levels and lowers physiological stress markers[50].

Cities with the highest compliance with the 3-30-300 rule tend to experience fewer effects from urban heat islands ($p < 0.001$, rho = −0.21; see Supplementary Fig. 5). This finding underscores the significant role that urban vegetation plays in reducing the exposure of urban populations to extreme land surface temperatures[51]. It highlights the substantial potential of trees and green spaces to mitigate urban heat in Europe. In this context, Iungman et al. (2023)[52] showed that increasing tree coverage to 30% would cool cities by a mean of 0.4 °C, which could prevent up to 1.84% of summer premature deaths. We acknowledge that achieving the target of 30% tree cover in densely built-up and populated cities presents significant challenges, as streetscapes alone are often insufficient. To overcome this, greening strategies must utilize other major land uses. Peri-urban forests should be prioritized for their positive impact on surface climate conditions and air quality[53]. Tree planting programs must also be expanded to private land and residential areas[54] which collectively represent a vast, often underutilized surface area for canopy growth. Greening of buildings (e.g., green roofs, green walls, and facade planting) should be a primary focus, particularly in dense urban areas where ground-level space is scarce, to add crucial vertical and horizontal canopy surfaces. Although trees offer numerous benefits, achieving 30% cover solely with trees in densely built environments may be difficult, especially in arid climates. In such scenarios, implementing more pocket-sized parks with a considerable tree component can be beneficial[22]. These parks, created from residual land, are accessible and appealing to local communities and easier to establish.

It is crucial to acknowledge that our findings must be interpreted with caution due to several methodological constraints that may influence the accuracy and applicability of the results.

A primary limitation of our assessment of Rule 3 (the 'three-tree' criterion) is the 100 m spatial grain of the GHS-BUILT-H and GHS-POP datasets. While the 3-30-300 rule is ideally evaluated from the micro-scale perspective of residential windows, the current lack of uniform, high-resolution pan-European datasets for building geometry and population distribution necessitates a coarser approach. This 100 m generalization may not capture micro-scale visibility nuances, such as individual trees visible through narrow gaps between buildings. To evaluate the impact of this generalization, we benchmarked our results against several high-resolution local studies[19–22]. Despite the differences in spatial grain, our findings are broadly consistent with these localized assessments (Supplementary Table 3). This alignment suggests that while our results represent a more generalized scale, they effectively capture the overarching spatial trends of urban forest visibility. This trade-off is essential for maintaining methodological consistency across the European continent; while local studies offer superior precision for individual cities, our approach provides the first harmonized, comparable baseline at a continental scale. Regarding Rule 30%, it is important to note the absence of a standardized neighborhood definition; therefore, we conducted a sensitivity analysis across various neighborhood sizes to evaluate the effect on the number of people in accordance with the 30% Rule (see Methods and Supplementary Fig. 7). Additionally, some studies incorporate all vegetated areas, including parks, whereas our analysis does not. Furthermore, it is crucial to acknowledge the rule's omission of blue spaces, which have been demonstrated to provide similar benefits to green spaces, including restoration, facilitating social interactions, and lowering psychological distress[55–57]. Finally, distances in the Rule 300

were calculated using the "as the crow flies" method, based on Euclidean measurements. While this approach is straightforward and efficient, it may not accurately account for physical barriers like buildings or roads without pedestrian crossings. To validate our method, we compared it with recent research by DG REGIO of the European Commission[58], which investigated access to urban green spaces within a 10-minute walking time along the street network. Our findings demonstrate a strong correlation between the two approaches ($p$-value « 0.01, $R^2 = 0.37$, see Methods and Supplementary Fig. 8), highlighting the robustness of our methodology. Our approach quantifies urban green space but does not address quality, a critical component for enhancing human well-being. Quality, encompassing environmental benefits but also aspects of relaxation and recreation, may significantly influence mental and physical health outcomes more than green space quantity alone[59,60]. Additionally, the 300-meter criterion does not account for usage patterns or preferences, as residents might favor more distant green spaces with preferred features over closer ones.

Despite the aforementioned limitations, our analysis marks a significant advancement, as it is, to our knowledge, the first to consistently map the 3-30-300 rule across more than 800 cities on a continental European scale. This study underscores the urgent need to foster a green transition that enhances urban livability and ensures more residents can benefit from well-integrated green infrastructure. The 3-30-300 rule offers a valuable framework that provides quantifiable metrics related to the visibility, coverage, and proximity of urban green spaces, enabling effective tracking of progress over time. This is particularly crucial in the context of the Nature Restoration Regulation (NRR), which mandates an increase in urban green spaces and tree canopy cover through innovative strategies, such as integrating green spaces into buildings and infrastructure. These strategies must prioritize accelerating canopy development by creating dense, diverse urban forests even in small plots, ensuring newly planted street trees have ample, uncompacted soil to mature fully, and transforming transport routes into shaded, continuous ecological corridors. Balancing competing land uses requires thoughtful planning and a commitment to preserving and expanding urban greenery. By leveraging our mapping of compliance with the 3-30-300 rule, European policymakers and urban planners can assess the current status of city dwellers' access to green areas, track progress over time, and develop scenarios for future improvements that are socially just and enhance equitable access to green spaces for all residents.

## Methods
### Selection of cities
Our analysis focused on European cities listed in the Urban Audit[61] dataset 2021, which follows the city definition by the Organization for Economic Cooperation and Development - European Commission (OECD-EC). This definition is based on criteria such as population density and local administrative boundaries[62]. From this original database, we specifically selected only the "City Core" areas, resulting in a total of 729 cities across 30 European Countries (EU27, Norway, Switzerland, and Iceland). To enhance our dataset, we also included 133 cities from the United Kingdom by accessing the 2020 Urban Audit dataset. This brought the total number of European cities considered in our analysis to 862.

### Rule 3
To assess the first rule, which specifies that every dwelling, educational facility, and workplace should have a view of 3 trees, here we used a special type of geospatial modelling, namely viewshed analysis[31,35,36], which delineates the area visible from a given observation point, taking into account the surrounding terrain features such as building height and tree cover estimates. The analysis was carried out using the QGIS 3.30.2 version and required three different datasets: distribution of

building height, estimates of tree cover count, and population distribution (GHSL-POP). Detailed information and the resolution of these layers are reported in Supplementary Table 1.

The spatial information about building height was retrieved from the Global Human Settlement Layers website[30]. The spatial raster dataset illustrates the distribution of building heights derived from global digital elevation models (DEMs) and filtered satellite imagery using linear regression. Two key DEMs were used: ALOS World 3D - 30 m (from 2006–2011) and Shuttle Radar Topography Mission 30 m (from 2000). The building height information was updated with shadow data from the 2018 Sentinel-2 satellite image, and the final product (GHS-BUILT-H) is provided at 100 m of resolution.

Similarly, the Global Human Settlement Layers provided the distribution of residential population, expressed as the number of people per cell[37]. Residential population estimates for 2020 were disaggregated from census or administrative units to 100 m resolution grid cells, informed by the distribution, volume, and classification of built-up as mapped in the Global Human Settlement Layer (GHSL) global layer per corresponding epoch. Therefore, it can be reasonably assumed that when the value of a grid cell is greater than zero, it indicates the presence of residential buildings where people actually reside.

To determine the proportion of areas containing at least three trees, we leveraged a recent methodology developed in Liu et al. (2023)[27] and Brandt et al. (2024)[28]. Individual tree count data were generated from PlanetScope, which is a constellation of 130+ nanosatellites with a spatial resolution of 3-meter and red, green, blue, and near-infrared bands, for the year 2019. The technical approach began with image preparation, where PlanetScope imagery was processed into custom 1° x 1° mosaic tiles, each comprising numerous scenes to ensure comprehensive coverage. A histogram-matching algorithm was applied to align these scenes with Landsat and Sentinel-2 data, creating a homogeneous mosaic. Image acquisition was strategically timed based on phenological windows derived from MODIS data. For deciduous regions, images were captured after the peak productivity of herbaceous vegetation, ensuring tree crowns were prominent. In evergreen areas, images were taken when trees had full green leaves, enhancing crown visibility. Given the importance of image quality, particularly sharpness, a blur kernel was used to assess each scene's sharpness directly after download. Scenes that did not meet a sharpness threshold of 0.23 were discarded, and alternative images were used until the required sharpness was achieved. The core of the tree detection method was a heatmap-based approach using convolutional neural networks (CNNs), which produced a confidence map where peaks indicated the centers of tree crowns. The models were trained using point labels for tree crown centers, with approximately 130,000 labels for PlanetScope manually verified using high-resolution imagery from Google Earth and Bing Maps. The models were trained over 2500 epochs with a learning rate that decreased linearly after the 2000th epoch. Multiple models were trained using different semi-random splits of the training data, ensuring robust performance across varied conditions. An ensemble of five models was then used to predict tree locations on the mosaics, averaging the results from these heatmaps. Local maxima were converted into point files representing tree crown centers, with associated confidence levels > 0.15 and minimal object spacing > 3 pixels. Starting from the identified tree crown centers, we generated a 100 m resolution raster of the number of trees for each city to ensure consistency with the resolution of the building (GHS-BUILT-H) and population (GHS_POP) raster data.

The first step of the viewshed analysis consisted of defining the number of observation points from which trees are supposed to be visible. Thus, for each city map, we created a set of viewpoints randomly sampled with a minimum spacing distance, namely, a point will not be added if there is an already generated point within this (Euclidean) distance from the generated location. We selected three minimum distance thresholds of 100 m, 250 m, and 500 m. Lower

thresholds could not be selected due to the spatial resolution of the 'obstacle layer' (building heights from GHS-BUILT-H) and the population distribution data (GHS-POP) that are only available at a 100 m grid resolution. The next step consisted of setting the radius of analysis, which is the maximum distance for visibility testing for the viewshed algorithm. Here we used two values, 100 m and 250 m, with the observer height set to 1.6 m. Then, we run the viewshed analysis to define as visible those portions of the landscape without obstructions (i.e., buildings or sloped terrain) from the building height layer, GHS-BUILT-H. Specifically, if an observation point exists within a cell that also contains trees, those trees are deemed "visible" based on proximity, unless the line of sight is blocked by an adjacent obstacle cell. This approach is a simplification of the "window-view" concept, but the high density of observation points ensures visibility is assessed from multiple angles, thereby reducing the likelihood of "dead zones" where trees might not be counted. Finally, visual exposure values from viewsheds generated from all sampling points are added, resulting in a continuous raster map of visual exposure. By combining the visible map obtained from our viewshed analysis with this new layer of grid cells containing at least three trees, we were able to map the pixels in each city that are visible from specified observation points and contain at least three trees. However, having a visible tree is not enough because we wanted to understand what is the amount of population that satisfies such a rule. Thus, to provide a more nuanced understanding of the rule's impact, we further analyzed the proportion of the urban population with access to these green spaces by combining these green pixels with the Global Human Settlement Layer (GHSL) to estimate the population living in visible areas ($\sum POP_{green}$), and then we normalize by the total population in the city ($POP_{city}$) to allow for comparison with the other two criteria of the 3-30-300 rule:

$$POP_{R3} = \sum POP_{green}/POP_{city} \qquad (1)$$

The population in urban areas with visibility of at least three trees has been calculated using four distinct parameter sets, specifically exploring three values for the minimum distance between viewpoints (d = 100 m, 250 m, 500 m) and two values for the radius of the viewshed analysis (r = 100 m, 250 m). The findings of this sensitivity analysis are illustrated in Supplementary Fig. 6. In the main text, we chose to present the results that offer the highest viewpoint density and the most refined spatial resolution, achieved with d = 100 m and r = 100 m. Further refinement of the spatial scale was limited by the resolution of the input datasets. In the viewshed analysis, the precision of the output is defined by the coarsest input layer, and to the best of our knowledge, at pan-European scale the most reliable building height data (GHS-BUILT-H) is currently restricted to a 100 m resolution. Therefore, using data on tree crown centroids and observation points distributed more densely (e.g., 10 m) against a 100 m obstruction block would not yield more accurate results; rather, it would imply a level of precision that the building height data cannot support. In particular, even if we use a set of viewpoints with higher density (e.g., 10 m of distance) we would still be limited by the radius of analysis, whose extent could be at minimum the size of the obstacle layer, and therefore 100 m.

## Rule 30

The second rule indicates that every neighborhood should have a 30% canopy cover. To examine the second rule, we utilized the Tree Cover Density database provided by PlanetScope. As for the previous rule, this dataset is obtained from individual tree count data generated from PlanetScope (spatial resolution of 3-meter) and aggregated to a 100 m resolution.

Due to the lack of a standardized definition of neighborhoods across cities, we define neighborhoods as 1 km x 1 km grid cells. This simplified definition is in line with the proxy for neighborhood defined in Konijnendijk (2023)[18] where a buffer of radius 500 meters was taken

to simulate people's daily neighborhood. To ensure that our focus remains on green-rich urban areas, we apply a threshold filter, retaining only those defined neighborhoods where the tree cover density is at least 30%. For the neighborhoods that meet this green density standard, we then proceed by summing the total population living in these 30% tree cover neighborhoods ($\sum POP_{in\_30}$) and then normalizing by the total cities' population ($POP_{city}$):

$$POP_{R30} = \sum POP_{in\_30} / POP_{city} \qquad (2)$$

To further assess the sensitivity of our results to the neighborhood definition as a $1 \times 1$ km grid, we calculated overall compliance with the 30% rule using three additional grids with resolutions of 500 meters, 250 meters, and 100 meters. Our sensitivity analysis (Supplementary Fig. 7) indicates that the choice of grid size does not show relevant differences in the overall compliance rates with the 30% Rule. For this reason, and in accordance with specific literature that often defines an egocentric neighborhood using a 500 m radius around each building[18,38–40,63], which we consider comparable to our $1 \times 1$ km approach, we decided to present the results evaluated at the 1 km grid (Fig. 1b).

## Rule 300

The third rule indicates that everyone should *live* within 300-m of a high-quality public green space of at least 1 ha in size[18,64]. The first step was then delineating those urban green spaces. To accomplish this, we started from the LUISA (Land Use-based Integrated Sustainability Assessment)[25] map, a spatial dataset developed by the European Commission's Joint Research Centre (JRC), which provides a detailed representation of land use and land cover across 39 countries in Europe at a high spatial resolution (50 m). From this map, we first extract the pixels corresponding to the Urban Area (pixels below 2000 as classification), then isolated areas classified as Urban Vegetation, Green Urban Areas, and Sport/Leisure Green within those urban areas. We then converted these areas into shapefile polygons using the polygonize function in QGis and kept only those that were at least 1 hectare in size, defining them as our green urban areas. We created 300-meter buffer zones around the delineated parks and green spaces to map the accessible green area for nearby residents. We then calculated the percentage of the city's population that lives within these zones to show which communities benefit from green space proximity. By examining the ratio of the population living within these buffer zones to the total population of the city, we gain insight into the proportion of residents who have ready access to such green spaces. This metric is crucial, as it informs urban planners and policymakers of the extent to which the city's population is served by accessible green spaces, a key factor in promoting urban livability, public health, and well-being. Finally, as per the other two criteria is normalized by the total city population.

$$POP_{R303} = \sum POP_{in\_300} / POP_{city} \qquad (3)$$

We further validated our approach by comparing it with recent work conducted by the DG REGIO of the European Commission[58] which utilized the Urban Atlas to assess access to urban green spaces. Their method defined proximity as a 400-meter walking distance to open and public areas, including green urban spaces. We compared our estimates of the population share meeting the R-300 rule with the proportion of urban center populations having access to at least 1 hectare of green urban areas within a 400-meter walk, as reported in the DG REGIO study. Our findings indicate a strong correlation between the two outputs (*p*-value « 0.01, R² = 0.37, see Supplementary Fig. 8), underscoring the robustness of our method. It is noteworthy that our results identify a smaller percentage of the population with access to green spaces. This discrepancy can be attributed to two main factors: (1) our use of a more restricted buffer distance (300 meters versus 400 meters) and (2) our focus on urban cores exclusively,

whereas the DG REGIO study encompasses the entire Functional Urban Area (FUA), which extends beyond the core city center.

## Sensitivity for the three rules together

Based on the various sensitivity analyses conducted for criteria R3 and R30%, we have identified different combinations that impact the overall fulfillment of the 3-30-300 rule (Supplementary Table S2). It is noteworthy that while the overall variations in rule fulfillment are relatively constrained, ranging from 9.3% to 13.5% of the urban population satisfying the criteria, variability is markedly more pronounced at the municipal scale. For instance, cities such as Essen (Germany), Gdansk (Poland), and San Sebastian (Spain) exhibit maximal differential ranges of approximately 20% (Supplementary Fig. 9). In the main text, we have prioritized presenting results that maximize adherence to the 3-30-300 rule. This condition is realized under criterion R3 with parameters D = 100 m and r = 100 m, and with the spatial grid size for neighborhood delineation in R30 set at 1 km x 1 km.

## Disposable Income

We use multiple sources for disposable income. The one characterized by the highest spatial resolution (200 × 200 m) is obtained from the National Institute of Statistics and Economic Studies (INSEE)[44] in France. For several years, INSEE has been disseminating indicators on taxable income or declared income (i.e., income before redistribution) of French households at all geographical levels permitted by respect for statistical confidentiality, from the district to the whole of metropolitan France, including a whole series of intermediate zonings: commune, employment zone, department, region.

Once we downloaded the dataset, we overlaid these 200 × 200 grids onto our spatial data concerning the fulfillment of the 3-30-300 rule (see Fig. 4a for the city of Lyon). Given the difference in spatial resolution between the two rasters, we defined the Green Space Category as the average value of compliance with the 3-30-300 rule. Therefore, when the Green Space Category is equal to 2, it indicates that all 100 m x 100 m grid cells within the 200 m x 200 m grid have an average value of 2. We repeat this for all 72 cities in France.

The other two datasets with a good spatial resolution of disposable income are Spain and Belgium. Disposable income data for Spain were sourced from the Instituto Nacional de Estadística[65]. These data were available at the levels of municipalities, districts, and census sections. We downloaded the Median Income Indicator for all census districts for the year 2018. As for France, we calculated the Green Space Category as the average level of compliance with the 3-30-300 rule across the various district extents. Disposable income data from Belgium were retrieved from Mikou et al. (2025)[43] since it was the country, along with France and Spain, with the highest spatial resolution for administrative units (-1 km2).

In our extended analysis, we have incorporated all other European cities from our database by utilizing the recent downscaling income product developed by Mikou et al. (2025)[43]. This product employs an innovative machine learning framework, providing a dataset at a $1 \times 1$ km resolution for the year 2015. The dataset expresses disposable income in terms of purchasing power parity (PPP) to account for variations in living costs across European countries. PPP rates are currency conversion rates that adjust for differences in price levels between countries. Where income data was missing for certain administrative units, they imputed values using the average income of neighboring units to ensure a complete and comprehensive dataset. Our approach mirrored the methodology used for France, Belgium, and Spain, where we compared income data with the Green Space Category. Nevertheless, the consistency between this product and our estimates from the three aforementioned countries is notable. Despite variations in climate, economy, and culture among cities, there is a discernible trend across our analysis: higher disposable income correlates with an increased likelihood of

residing in greener neighborhoods (Fig. 4B and Supplementary Fig. 2). Additionally, we provide a localized comparison with the city of Lyon to highlight the similarities in trends (see Supplementary Fig. 10). This analysis underscores the broader pattern observed across diverse European contexts.

## Reporting summary
Further information on research design is available in the Nature Portfolio Reporting Summary linked to this article.

## Data availability
The 3-30-300 spatial maps generated in this study have been deposited in the Zenodo database[66]. The data supporting the findings of this study are entirely derived from publicly available resources. The PlanetScope tree cover products (2019) were accessed via the published literature link (https://www.nature.com/articles/s41893-024-01356-0). The GHSL-BUILT-H building height (2018) and GHSL-POP population density (2020) datasets are accessible through the GHSL Copernicus portal (https://human-settlement.emergency.copernicus.eu/datasets.php). The LUISA Base Map (2018), used specifically for urban green vegetation, is available via the JRC data portal (https://data.jrc.ec.europa.eu/dataset/51858b51-8f27-4006-bf82-53eba35a142c). Finally, the extent of the City Core boundaries for EU (2021) and UK (2020) cities was sourced from the Urban Audit via Eurostat (https://ec.europa.eu/eurostat/web/gisco/geodata/statistical-units/urban-audit).

## Code availability
QGIS (version 3.30.2) was used for the present geospatial analysis, and RStudio (version 1.4.1717) was used for the statistical analysis. All the source codes are available upon request to the corresponding author.

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

## Acknowledgements

The views and opinions expressed in this article are those of the authors and do not necessarily reflect the official position of the European Commission.

## Author contributions

L.E.B, L.F., and M.V.V conceived the study. L.E.B. ran the spatial analysis for the estimate of the 3-30-300 rule in European cities. M.B and S.L. carried out the analysis on individual tree counts from Planet Scope. L.E.B, L.F., J.M., and M.V.V. interpreted the results and wrote the manuscript with contributions from all authors.

## Competing interests

The authors declare no competing interests.
