## [Transparent Peer Review file · Nature Communications]

Assessing European Cities with the 3-30-300 rule underscores the need for enhanced urban greening efforts

Corresponding Author: Dr Leonardo Bertassello

Version 0:

Reviewer comments:

Reviewer #1

(Remarks to the Author)

This article addresses crucial issues related to quality of life in European cities, focusing on access to urban green spaces, one of the most important factors. The research was conducted based on the 3-30-300 concept, which has recently become popular due to its transparent criteria. The research was clearly described, with a fairly detailed account of the methodology and results obtained, which made the entire process transparent and comprehensible. While I appreciate the significance of the overall research presented, I have a few comments regarding the adopted methodology and a few minor technical observations. They are as follow:

1. The chapter called 'Main' (perhaps 'Introduction' would be more appropriate) lacks references to other concepts related to the study of quality of life in cities, including those that consider access to green spaces, such as 15-minute cities and biophilic cities. The 3-30-300 concept is not the only one in this field. It would be worthwhile justifying why this particular concept was chosen for this research.
2. Line 34-35 – there is a lack of references after “to meet UN Sustainable Development Goal 11 Target 7”.
3. Figure 1, Figure 2 - It would be helpful if the drawings (as well as titles) could be enlarged, as the descriptions in the legend are very difficult to read.
4. Results – views 3 trees – “Half of the European urban population views 3 trees” – according to the methodology used, I have little confidence in these conclusions. I will explain this further in the comments on the methodology.
5. Results – 30% tree cover - According to the method used, the results of the 30% tree cover rule could be even better than presented because areas smaller than 1 km² are removed. It would be useful to add such a comment. On the other hand, calculating the population with such a low level of resolution seems to introduce a rather high degree of generalisation.
6. Line 114 – Warszawa is Warsaw in English.
7. Results - Line 123 and subsequent lines – “Fifty percent of European city residents enjoy proximity to green spaces” – in my opinion, this term is too strong. Green spaces are not evenly distributed in virtually any city. The essence of the green city concept is to examine this phenomenon locally in order to identify areas where green spaces have been developed well and areas where they have not. This sentence only refers to the distance from the park.
8. Line 228 – “Opportunities and challenges of the 3-30-300 rule in urban planning” - The title does not accurately reflect the content of the chapter. It primarily refers to the opportunities and challenges of developing green areas in urban planning, rather than the 3-30-300 rule. The main challenge with this concept is how to quantify the rules and transform them into measurable parameters.
9. Line 275-276 – “Nature Restoration Regulation (NRR)” – there is a lack of reference 13.
10. Methods - The selection criteria for the cities is not entirely clear to me. Was the city with the highest population density chosen?

11. Methods – Rule 3 – This part of the research is its biggest flaw. Selecting measurement points at 500-metre intervals greatly simplifies the analysis. In my opinion, the authors mistakenly applied the strictly local 3-30-300 rule concept to global analyses. The 3-30-300 concept is a simple and useful tool for examining our immediate surroundings in great detail. But the main assumption of the concept is “to bring trees and nature all the way into people’s neighbourhoods, streets, and on their doorsteps in order to capitalise on their many benefits”. However, simplifying to a resolution of 500 metres renders it very general, detached from the underlying ideas and assumptions. Assuming the concept that each resident should see three trees from their apartment, the study should include every house or even every floor of a building. In my opinion, a resolution of 500 metres between measurement points produces unreliable results.

Moreover, the 3-30-300 concept does not specify the distance at which residents should plant these trees. The 250-metre rule contradicts Konijnendijk’s assumption that trees in close proximity are visible. A distance of 250 metres does not allow us to see greenery from our windows.

Furthermore, what was the height of the observation point? This significantly affects the visibility of trees. Someone on the first floor might see something completely different to someone on the fifth or tenth floor.

12. Methods – line 336 – 100x100 grid – 100m, 100 km? What size is the grid?

13. Methods – Rule 30 – The 1 km grid eliminates several smaller parks from the analysis. Once again, we are moving away from Konijnendijk’s idea of locality.

14. Methods - Rule 300 – line 376 – The resolution is 50 m. This part of the research is at a completely different level to the 3-trees rule (50 m versus 500/250 m).

15. Lines 398-410 - This part is more of a discussion than a method.

16. Methods - Disposable Income - The data being compared here have different resolutions. I understand that it is problematic to have the same data for different countries. However, the disadvantages of comparing data at different levels of detail must be clearly explained.

In summary, we believe that the main strengths of this article are its focus on the important social issue of quality of life in cities and its analysis of access to green spaces using the 3-30-300 concept, which has recently gained popularity due to its apparent simplicity. The main drawback of this article is that it uses the 3-30-300 concept for European-scale research, utilising relatively low-detail data, whereas the essence of this concept lies in its local application (e.g. proximity to green spaces and residences). Therefore, in my opinion, the results obtained, mainly regarding the three-tree visibility requirement, raise serious doubts. Direct tree visibility is measured using parameters that are larger than the accessibility to large parks (measuring points every 500 m and visibility of 250 m, while access to the park in a distance of 300m). These are not comparable principles and should not be analysed together.

However, in order to preserve the article’s value, I propose rewording it to clarify the problems and inaccuracies arising from the aforementioned flaws. At the same time, we should avoid making overly strong, unequivocal statements that are not supported by the results obtained. The description should reflect the fact that the results are not entirely precise.

Reviewer #2

(Remarks to the Author)

Reviewer #3

(Remarks to the Author)

This is a straightforward and well-written paper, offering what is likely the largest 3-30-300 study to be published in terms of its spatial scale, offering novelty in its coverage of the entirety of Europe’s urban regions. It conclusively supports the insights offered by past studies: urban residents have poor access to green space, and this is a consistent problem across very diverse urban regions. I appreciated the examination of wealth and aridity as mediating factors, this was a valuable progression on the base message. The datavis was also a strength.

I really enjoyed reading this paper and had very few concerns with the Main section; these are detailed below.

However, on reaching the Methods I came to understand how coarse this analysis is, and also felt that some of the care in the previous sections was missing, with a few key decisions in the method lacking justification. All three ‘tests’ for the 3-30-300 are conducted at a very coarse spatial scale, with quite coarse data. Canopy data is at 10m resolution, neighbourhoods for canopy assessment are 1km x 1km blocks and viewsheds are based on ‘observation points’ at very low density per square kilometre. Tree point locations are from 2019. Park access was also calculated as a ‘crow flies’ distance using

buffers rather than walking distance along footpaths. Some simplification of methods is understandable given the scale of the study, but my concern is that the research must remain consistent with the intent of the 3-30-300 rule, which aims to localise access to urban nature and pull focus to what is accessible from buildings. The methods applied here feel more aggregated, with no stage of the method using buildings as a direct unit of testing.

The 3 test's viewshed method was difficult to follow and may benefit from some interpretive visuals. Why are you measuring which 100m x 100m pixels are visible within viewsheds that are >500m apart? This does not appear to be a suitable proxy for testing sightlines from buildings. It was also unclear how viewsheds were selected, and why 100m x 100m pixels were opted for. My concern is that this does not really indicate whether the buildings have views to trees. This highlights a key challenge in the method. The 3-30-300 method really looks at buildings, not population units. It seems that use of population units as the basis of the method produces significant issues of aggregation. Buildings, by contrast, can be tested individually, as has been demonstrated in past large 3-30-300 studies.

1km neighbourhood blocks for the 30 test are also not justified and this part of the method seems an especially severe deviation from the intent of the metric. At this scale, individual buildings could be hundreds of metres from significant patches of canopy, while still passing the '30' test. The authors are correct that 'neighbourhood' is not clearly defined – a key issue with the method itself – but this does not necessarily imply that a highly aggregated spatial scale is appropriate. I would strongly recommend a finer scale for this step of the analysis, as well as a clear justification of the scale selected. One option is to select a distance at which thermal effects of canopy – or perhaps a selected health benefit – are detectable.

The 300 test's use of buffers is less of a serious problem but must be better signposted in the Main text (e.g. at line 126,127 the term 'buffer' could have been inserted as a cue to the method used). A discussion of how this serves to overestimate access would also be appropriate, ideally backed by a short sample analysis demonstrating the scale of overestimation/uncertainty this introduces to the study relative to more precise network-based routing analysis methods. Overall I am quite concerned that the use of population has broken the method. Ideally this study would carry out building-scale tests, then link these results to population by making a statistical argument around how many people each building might represent. This avoids the need to use very aggregated methods which introduce many of the problems that 3-30-300 was conceived to avoid.

Minor issues in the Main text:

Line 76 – What's LUISA?

85 – explanation of views is unclear.

Figure 3 – this is nice but a little confusing. Is the x-axis 'share of population'... the share of the population passing the 3-30-300 test? Please annotate this axis fully. This confusion is partly sparked by line 190-191 which state the boxes show the IQR of income values. I also suggest you change the aridity quartiles to abcd instead of 1234 so that it's easy to discern what '2-4' means (at the moment it's tricky because one is not sure whether aridity is first or second and this takes a minute to confirm).

Figure 3 also does appear to show that aridity is a major driver of low 3-30-300 achievement – the box for 3-2 seems to indicate that in the third income quartile, the lowest humidity cities have much lower achievement. The text at lines 177-180 appears to deny this.

203-210. Please justify why a more granular analysis was necessary.

231 – I object to the use of 'inevitably'. Surely there are areas where green access is good even in areas which have urbanised. This undermines a core message that is important in this field of research: urbanisation does not need to mean huge losses of greenery. The problem is one of design; we have agency in this. The term inevitably implies that infill development is an unquestionable force of 'greying'. To date it mostly has been, and perhaps this is a better way to acknowledge the issue, without creating a sense of helplessness.

241-257. These are good arguments but streets are a minority of urban space, often 30-40%. What other land uses can play a role in greening? Streets alone won't get us to 30.

265-269. As with my comment regarding inevitability above: the argument that 30% in urban areas is unfeasible is not supported by your evidence. 30% cover is more difficult to deliver in urbanised areas but 'spatial and environmental limitations' are the product of design decisions; if an urban area has no space for trees then it has been designed badly.

These are not feasibility problems. Indeed, the street retrofit ideas you indicate in 241-257 are promising remedies to these more dense areas, especially as these more walkable dense areas do not need to cater to drivers as generously.

Finally – your title is very punchy, but it uses the term 'green space'. I think many readers will interpret this as 'parks', which was my initial impression too. A less ambiguous term may help avoid this misconstrual of the study's focus.

Version 1:

Reviewer comments:

Reviewer #1

(Remarks to the Author)

Thank you to the authors for addressing the proposed revisions and carefully responding to the comments raised. The changes introduced in many places have clearly improved the clarity of the research and its presentation. Numerous parameters have been explained, formulations have been refined, the argumentation has been better structured and key aspects of the methodology have been clarified. Several previously ambiguous sections have been significantly clarified. Consequently, I now have a much better understanding of the research, and the article is more coherent and easier to follow.

While the explanations and arguments provided by the authors are sufficient for most of the comments I raised, two issues still require further refinement in my opinion.

The first issue concerns Rule 1 (the visibility of three trees). The additional explanations provided by the authors greatly

enhance the reader's understanding of the research process, which appears logical and interesting. However, I remain concerned about the level of detail in the analysis and the conclusions drawn from it. For instance, given the tree centroids and observation points, I do not understand why the authors first generalised the tree information by counting trees within a 100 m grid cell and only then determined their visibility. Performing this operation prior to generalising the tree data would have yielded more precise results.

Moreover, I do not understand how grid cells containing obstacles that block the visibility of trees were taken into account within the adopted workflow. In particular, in the case of 100 m axis, in particular, the cells obstructing visibility may be the same as those containing the three trees. What, then, is the outcome of the analysis in such cases? What happens when the cell that blocks visibility is the same cell where the observation point falls?

In my opinion, the results obtained in this rule are highly generalised (and thus unreliable) for another reason. The spacing of observation points at 100-metre intervals is too coarse; within a radius of 100 metres, several buildings and other obstacles may be present, leading to very different local visibility conditions. Using such large distances to assess tree visibility moves the analysis away from the core principle of the 3-30-300 concept: that greenery should be in residents' immediate surroundings. Using data on tree crown centroids and observation points distributed more densely (e.g. 10m) would make it possible to obtain fairly precise results that capture local conditions in a very straightforward manner, which is the essence of Konijnendijk's concept. Moreover, these assumptions strongly influence the general results of the three-tree criterion assessment, yet they are addressed with only a single sentence of commentary in the limitations section.

The second issue relates to the language used to describe the conclusions. I raised this point in my first review. Many of the formulations, especially given the high level of generality of the analyses, are overstated. In other words, the conclusions drawn are too strong and radical given the level of detail in the data used and the results obtained. The authors use overly strong wording which is often not fully justified. Such statements might be appropriate if the level of detail in the data corresponded to the assumptions of the concept. However, since this is not the case, the research results require a more critical and cautious interpretation.

E.g.:

"Less than 15% of the population in European cities..."

"A quarter of city-dwellers in Europe..."

"Half of the European urban population..."

If the study only analyses 800+ cities in Europe, whereas there are around 2,000 cities in Germany alone, then it still only covers a portion of the population in European cities. While it does represent the populations of the largest and most urbanised cities, it does not represent the entirety of the urban population.

In summary, while acknowledging the amount of work involved, I recommend recalculating the three-tree criterion using observation points that are more densely distributed. This would substantially increase the value of the article and ensure that the conclusions drawn are well-founded.

Reviewer #2

(Remarks to the Author)

Reviewer #3

(Remarks to the Author)

Thanks for appropriately acknowledging the limitations of this study, and producing the sensitivity analyses that show relatively low median differences. I feel this is now ready for publication, and will likely cite it shortly. Thanks for an important piece of work.

Version 2:

Reviewer comments:

Reviewer #1

(Remarks to the Author)

I have carefully reviewed the revised paper currently introduced by the Authors. I am pleased to accept most of these changes, particularly as they significantly refine the terminology used to describe the research and its findings. This indicates a shared understanding regarding a crucial aspect: the appropriate mode of scientific discourse—specifically, using language that accurately reflects the factual basis of the study. In my assessment, this provides necessary clarity to the results section and establishes a sound framework for their empirical interpretation.

Furthermore, I appreciate the additional details concerning the study's limitations, specifically those regarding the visibility of the three trees. These explanations are pertinent and essential, as they address the deficiencies and imperfections within both the conducted work and the datasets employed. Consequently, they facilitate a more accurate interpretation of the

results obtained. Maintaining a critical perspective on one's own research is vital for ensuring scientific objectivity. Research is not required to be flawless, and researchers frequently encounter data constraints. Such constraints do not constitute errors; rather, they represent common methodological limitations. An actual error lies in drawing overreaching conclusions that are not supported by the data.

I maintain the position that local-scale studies, given their granular nature, should not be based on excessively generalized data, as this introduces significant margins of error. The argument that one must utilize available data in the absence of more appropriate sources is unconvincing; the principle of data adequacy should always be the primary consideration.

Nevertheless, considering the substantive value of this research, its compelling methodological approach, and its significance for both the quality-of-life research community and those advancing spatial analysis methods in this field, I recommend this paper for publication. I would only further suggest incorporating the explanations provided in the responses to my previous comments—specifically regarding grid cells containing obstacles that block the visibility of trees (R1C3)—into the data description or methodology section of the manuscript.

Reviewer #2

(Remarks to the Author)

Editor's note: Translated comments.

I co-reviewed this manuscript with one of the reviewers who prepared the reports mentioned. This is part of a Nature Communications initiative to facilitate training in peer review and ensure appropriate recognition for young researchers who co-review articles.

Współrecenzowałem ten rękopis z jednym z recenzentów, którzy przygotowali wymienione raporty. Jest to część inicjatywy Nature Communications, mającej na celu ułatwienie szkoleń w zakresie recenzji naukowej oraz zapewnienie odpowiedniego uznania dla młodych badaczy, którzy współrecenzują artykuły.

Response to reviewers for the manuscript

Less than fifteen percent of European city residents meet the 3-30-300 rule for green space visibility, canopy, and proximity

L.E. Bertassello ^{1*}, M. van der Velde ¹, J. Maes ², S. Liu ³, M. Brandt ³, L. Feyen ¹

1. European Commission, Joint Research Centre, Ispra, Italy
2. European Commission, Directorate-General for Environment, Brussels, Belgium
3. Department of Geosciences and Natural Resource Management, University of Copenhagen, Copenhagen, Denmark

* Corresponding Author: Leonardo Enrico Bertassello, leonardo.bertassello@ec.europa.eu,
ORCID: <https://orcid.org/0000-0001-5168-2142>

Note: the answer to reviewers' comments are highlighted in red, and the references to the text refers to the updated and clean version of the manuscript.

Reviewer #1 (Remarks to the Author):

This article addresses crucial issues related to quality of life in European cities, focusing on access to urban green spaces, one of the most important factors. The research was conducted based on the 3-30-300 concept, which has recently become popular due to its transparent criteria. The research was clearly described, with a fairly detailed account of the methodology and results obtained, which made the entire process transparent and comprehensible. While I appreciate the significance of the overall research presented, I have a few comments regarding the adopted methodology and a few minor technical observations. They are as follow:

1. The chapter called 'Main' (perhaps 'Introduction' would be more appropriate) lacks references to other concepts related to the study of quality of life in cities, including those that consider access to green spaces, such as 15-minute cities and biophilic cities. The 3-30-300 concept is not the only one in this field. It would be worthwhile justifying why this particular concept was chosen for this research.

We appreciate the reviewer's suggestions and have incorporated additional concepts related to urban quality of life, as indicated in Lines 36-41 and 266-270 of the revised manuscript:

“In the context of a rapidly urbanizing world, concepts like the biophilic cities¹⁰ or smart sustainable cities¹¹ urge urban planners to facilitate daily interactions with nature, promoting environmental awareness and stewardship. These cities are envisioned to be sustainable and resilient, strengthening their social and physical capacity against future shocks such as climate change, natural disasters, and economic uncertainties.”

And

“A key solution to achieving this balance is the transformation of urban mobility⁴⁶, a solution also advocated by the recent concept of the 15-minute city^{47,48}, which contends that cities can function more effectively, equitably and environmentally if essential services and key amenities are within 15 minutes by non-polluting forms of transport, such as walking or cycling.”

The decision to focus on the 3-30-300 concept in our research stems from its provision of a quantifiable, straightforward guideline that is universally applicable and measurable (Line 58: *“the rule is easy to remember, and as we will show, relatively straightforward to implement, monitor, and evaluate.”*).

Furthermore, similar to the concepts mentioned, adopting the 3-30-300 rule enables urban planners to systematically work towards creating healthier, more sustainable, and resilient cities (see Line 62):

“The 3-30-300 rule provides a set of objectives, universal benchmarks for cities to determine how they fare on urban green space visibility, coverage and proximity, as well as where to target investments for the future.”

Additionally, it facilitates monitoring the green status of cities as more data becomes available (Lines 332-340):

“The 3-30-300 rule offers a valuable framework that provides quantifiable metrics related to the visibility, coverage, and proximity of urban green spaces, enabling effective tracking of progress over time. This is particularly crucial in the context of the European Union’s Nature Restoration Regulation (NRR), which mandates an increase in urban green spaces and tree canopy cover through innovative strategies, such as integrating green spaces into buildings and infrastructure. These strategies must prioritize accelerating canopy development by creating dense, diverse urban forests even in small plots, ensuring newly planted street trees have ample, uncompacted soil to mature fully, and transforming transport routes into shaded, continuous ecological corridors.”

These considerations underscore the choice of the 3-30-300 concept due to its clear, actionable metrics and its potential to significantly enhance urban living standards.

2. Line 34-35 – there is a lack of references after “to meet UN Sustainable Development Goal 11 Target 7”.

Thanks for noticing, we added the appropriate reference [9]: *“Assembly, U. G. (2015). Transforming our world: the 2030 Agenda for Sustainable Development”*

3. Figure 1, Figure 2 - It would be helpful if the drawings (as well as titles) could be enlarged, as the descriptions in the legend are very difficult to read.

Thanks for the suggestion, we hope the revised Figures 1 and 2 provide enhanced clarity. In Figure 1, we have removed the previous PDF plots, as they were not discussed in the main text, and directed readers to Extended Figure 1, which serves as a surrogate for this information at the city scale.

Figure 1: (a)-(b)-(c) Spatial representation of the fulfillment of each single rule of the 3-30-300 guideline for 862 European cities. The circle size is proportional to the total city population, while the discrete color coding represents the proportion of the city's population that meets each rule's requirement. The lower panels.

Figure 2: (a) Spatial representation of the fulfillment of the 3-30-300 guideline for 862 European cities. The circle size is proportional to the total city population, while the discrete color coding represents the proportion of the city's population that meets each rule's requirement. (b) High-resolution analysis for a sample of 6 cities, showing the number of fulfilled rules at a 100x100 m grid cell level. This finer-scale representation reveals the intra-urban variability in guideline fulfillment, highlighting areas where multiple rules are met or not.

4. Results – views 3 trees – “Half of the European urban population views 3 trees” – according to the methodology used, I have little confidence in these conclusions. I will explain this further in the comments on the methodology.

We appreciate the reviewer's feedback regarding the section on viewing three trees. In response to comment R1C11 (that is referred to here by R1), we have substantially revised this section to enhance clarity and confidence in our methodology. We hope that the updated content effectively addresses such concerns and provides a more robust understanding of our approach.

5. Results – 30% tree cover - According to the method used, the results of the 30% tree cover rule could be even better than presented because areas smaller than 1 km² are removed. It would be useful to add such a comment. On the other hand, calculating the population with such a low level of resolution seems to introduce a rather high degree of generalisation.

We appreciate the reviewer's feedback highlighting the limitations and potential improvements in our methodology regarding the 30% tree cover rule. In response, we have

revised the methods section (see Lines 466-473 and Figure S7) to include a sensitivity analysis using three additional grid sizes as proxy for our neighbourhood definition: 100m, 200m, and 500m.

“To further assess the sensitivity of our results to the neighborhood definition as a 1 x 1 km grid, we calculated overall compliance with the 30% rule using three additional grids with resolutions of 500 meters, 250 meters, and 100 meters. Our sensitivity analysis (Figure S7) indicates that the choice of grid size does not show relevant differences in the overall compliance rates with the 30% Rule. For this reason, and in accordance with specific literature that often defines an egocentric neighborhood using a 500 m radius around each building^{18,38,39,40,63}, which we consider comparable to our 1 x 1 km approach, we decided to present the results evaluated at the 1 km grid (Figure 1b).”

We have also added a dedicated paragraph to the Limitations section (Lines 305-313) detailing the methodological considerations of the 30% rule.

“Regarding Rule 30%, it is important to note the absence of a standardized neighborhood definition; therefore, we conducted a sensitivity analysis across various neighbourhood sizes to evaluate the effect on the number of people in accordance with the 30% Rule (see Methods and Figure S7). Additionally, some studies incorporate all vegetated areas, including parks, whereas our analysis does not. Furthermore, it is crucial to acknowledge the rule's omission of blue spaces, which have been demonstrated to provide similar benefits to green spaces, including restoration, facilitating social interactions, and lowering psychological distress^{55,56,57}.”

These adjustments aim to provide a more comprehensive and reliable assessment of our findings.

Figure S7: Sensitivity analysis of the proportion of the population in cities complying with the 30% rule across different grid sizes (i.e., varying neighborhood sizes). Each point represents a city in our database, with black horizontal dashed lines indicating the average compliance with the rule. At the top, we provide an example from the city of Thessaloniki, Greece, where black pixels denote "neighborhoods" achieving 30% tree cover.

6. Line 114 – Warszawa is Warsaw in English.

Thank you for noticing, we edited accordingly in Line 124.

7. Results - Line 123 and subsequent lines – “Fifty percent of European city residents enjoy proximity to green spaces” – in my opinion, this term is too strong. Green spaces are not evenly distributed in virtually any city. The essence of the green city concept is to examine this phenomenon locally in order to identify areas where green spaces have been developed well and areas where they have not. This sentence only refers to the distance from the park.

We thank the reviewer for the suggestion and have changed the subtitle to “Over half of European city residents live within 300 meters of a park”, which more correctly represents what we assess.

8. Line 228 – “Opportunities and challenges of the 3-30-300 rule in urban planning” - The title does not accurately reflect the content of the chapter. It primarily refers to the

opportunities and challenges of developing green areas in urban planning, rather than the 3-30-300 rule. The main challenge with this concept is how to quantify the rules and transform them into measurable parameters.

We thank the reviewer for the suggestion. In the revised manuscript, we have divided the section into two parts: "Opportunities for urban green space development" and "Challenges and limitations." We believe that this new structure more accurately reflects the content of the chapter.

9. Line 275-276 – "Nature Restoration Regulation (NRR)" – there is a lack of reference 13.

Thanks for noticing, we added the appropriate reference [16]: "*Regulation (EU) 2024/1991 of the European Parliament and of the Council on nature restoration and amending Regulation (EU) 2022/869 (Text with EEA relevance) (Official Journal of the European Union, 2024).*"

10. Methods - The selection criteria for the cities is not entirely clear to me. Was the city with the highest population density chosen?

We apologize for the lack of clarity. We did not choose cities based on having the highest population density. Instead, the selection was guided by the predefined designation within the Urban Audit dataset, which encompasses central urban areas as defined by the OECD-EC guidelines. This approach ensures a comprehensive representation of urban centers across the specified European countries, rather than focusing solely on population density metrics. We address this in Lines 348-355 of the revised manuscript:

"Our analysis focused on European cities listed in the Urban Audit⁶¹ dataset 2021, which follows the city definition by the Organization for Economic Cooperation and Development - European Commission (OECD-EC). This definition is based on criteria such as population density and local administrative boundaries⁶². From this original database we specifically selected only the "City Core" areas, resulting in a total of 729 cities across 30 European Countries (EU27, Norway, Switzerland and Iceland). To enhance our dataset, we also included 133 cities from the United Kingdom by accessing the 2020 Urban Audit dataset. This brought the total number of European cities considered in our analysis to 862."

11. Methods – Rule 3 – This part of the research is its biggest flaw. Selecting measurement points at 500-metre intervals greatly simplifies the analysis. In my opinion, the authors mistakenly applied the strictly local 3-30-300 rule concept to global analyses. The 3-30-300 concept is a simple and useful tool for examining our immediate surroundings in great detail. But the main assumption of the concept is "to bring trees and nature all the way into people's neighbourhoods, streets, and on their doorsteps in order to capitalise on their many benefits". However, simplifying to a resolution of 500 metres renders it very general, detached from the underlying ideas and assumptions. Assuming the concept that each resident should see three trees from their apartment, the study should include every house or even every floor of a building. In my opinion, a resolution of 500 metres between measurement points produces unreliable results.

Moreover, the 3-30-300 concept does not specify the distance at which residents should plant these trees. The 250-metre rule contradicts Konijnendijk's assumption that trees in

close proximity are visible. A distance of 250 metres does not allow us to see greenery from our windows. Furthermore, what was the height of the observation point? This significantly affects the visibility of trees. Someone on the first floor might see something completely different to someone on the fifth or tenth floor.

We appreciate the reviewers for highlighting these points and apologize if our previous methodological description lacked clarity and caused confusion.

Firstly, we utilized a resolution of 100 meters for the three primary raster layers in our analysis: building height (GHS-BUILT-H), population distribution (GHS_POP), and tree count. As specified in Lines 381-385 and 406-409, the tree count was initially generated using 3 x 3 meter grid cells and then aggregated for consistency with the other geospatial data layers:

“To determine the proportion of areas containing at least three trees, we leveraged a recent methodology developed in Liu et al., (2023)²⁴ and Brandt et al., (2024)²⁵. Individual tree count data were generated from PlanetScope, which is a constellation of 130+ nano-satellites with a spatial resolution of 3-meter and red, green, blue and near-infrared bands, for the year 2019.”

“Starting from the identified tree crown centers, we generated a 100 m resolution raster of the number of trees for each city to ensure consistency with the resolution of the building (GHS-BUILT-H) and population (GHS_POP) raster data”.

Secondly, we conducted a sensitivity analysis on two parameters of the viewshed analysis: the minimum spacing distance (100 m, 250 m, and 500 m) and the visibility analysis radius (100 m and 250 m) (see Lines 412-424, 435-443, and Figure S6).

“The first step of the viewshed analysis consisted in defining the number of observation points from which trees are supposed to be visible. Thus, for each city map we created a set of viewpoints randomly sampled with a minimum spacing distance, namely a point will not be added if there is already a generated point within this (Euclidean) distance from the generated location. We selected three minimum distance thresholds of 100 m, 250 m, and 500 m. Lower thresholds could not be selected due to the spatial resolution of the analyzed raster layers (i.e., 100 m). The next step consisted in setting the radius of analysis, which is the maximum distance for visibility testing for the viewshed algorithm. Here we used two values 100 m and 250 m, with the observer height set to 1.6 m. Then, we run the viewshed analysis to define as visible those portions of landscape without obstructions (i.e., buildings or sloped terrain) from the building height layer, GHS-BUILT-H. Finally, visual exposure values from viewsheds generated from all sampling points are added, resulting in a continuous raster map of visual exposure.”

“The population in urban areas with visibility of at least three trees has been calculated using four distinct parameter sets, specifically exploring three values for the minimum distance between viewpoints ($d = 100\text{m}, 250\text{m}, 500\text{m}$) and two values for the radius of the viewshed analysis ($r = 100\text{m}, 250\text{m}$). The findings of this sensitivity analysis are illustrated in Figure SX2. In the main text, we chose to present the results that offer the highest viewpoint density and the most refined spatial resolution, achieved with $d = 100\text{m}$ and $r = 100\text{m}$. Further

refinement of the spatial scale was constrained by the limitations in the resolution of the input datasets. With this parameter combination, 46.7% of the population meets the criteria specified by rule R3.”

Figure S6. Sensitivity Analysis of Population Visibility of Trees (Rule R3). The scatter plot illustrates the range of estimated population shares meeting the Rule R3 criteria (visibility of at least 3 trees) across 862 European cities, based on four distinct parameter sets for the viewshed analysis. The analysis varied the minimum distance between viewpoints (d : 100m, 250m, 500m) and the viewshed radius (r : 100m, 250m). The final parameter set chosen for the main text results— $d = 100\text{m}$ (highest viewpoint density) and $r = 100\text{m}$ (most refined spatial resolution)—yielded the maximum population share meeting the criteria, with an estimated 46.7% of the population having visibility of at least three trees. This selection offers the highest feasible spatial resolution, constrained by the input dataset limitations.

While we acknowledge that the "3-30-300 concept does not specify the distance at which residents should plant these trees," various studies have made assumptions about visibility distance when calculating R3. For example, Cimburova and Blumentrath (2022) conducted viewshed modeling for exposure distances ranging from 50 to 300 meters. Other researchers define tree visibility as the presence of trees within a buffer with varying distance thresholds, from 15 m to 100 m (Nieuwenhuijsen et al. 2022; Battisti et al. 2023; Daland, 2023; Owen et al., 2024). We added such information in L. xxx of the revised manuscript:

“Konijnendijk (2023)¹⁸ suggests that every resident should have a view of three trees from their dwelling, educational facility and workplace. However, defining this visibility is challenging. Various studies employ different methods: some use surveys and window-view analyses to assess tree visibility from households^{31,32}, others utilize computer vision to

quantify street greenery³³, and some define tree visibility using buffer zones with distance thresholds^{19,20,34}. Here, to assess the visibility of three trees, we used viewshed analysis^{31,35,36} (see Methods). First, we mapped areas visible from specific observation points, accounting for obstructions like building height and surrounding terrain features. We then integrated these maps with high-resolution (3 m) tree count data from PlanetScope^{27,28} (see Methods) to focus on raster cells containing at least three trees. Finally, we applied this layer of visual green exposure to the Global Human Settlement Layer Population³⁷ (GHS-POP) data to estimate the population residing in areas with visibility of at least three trees.”

Given the scope of our analysis—which is to map the 3-30-300 rule consistently across more than 800 cities on a continental European scale—we are confident that the 100-meter threshold aligns with limits established in the literature. Due to the limitations inherent in the data layers we used, we were unable to conduct this analysis at a finer scale. These constraints are documented in both the methods section (see Section **Rule 3**) and the main text (see Section **Challenges and limitations**). Nonetheless, this remains a limitation that future research and advancements in satellite datasets could address.

Finally, we selected an observation point height of 1.6 meters. We recognize that residents on different floors may have varying visibility, but unfortunately, there is no available database that maps population distribution across different floors at a pan-European scale with a 100-meter resolution.

12. Methods – line 336 – 100x100 grid – 100m, 100 km? What size is the grid?

Thanks for noticing, we edited in Lines 406-409 as:

“Starting from the identified tree crown centers, we generated a 100 m resolution raster of the number of trees for each city to ensure consistency with the resolution of the building (GHS-BUILT-H) and population (GHS_POP) raster data”.

13. Methods – Rule 30 – The 1 km grid eliminates several smaller parks from the analysis. Once again, we are moving away from Konijnendijk's idea of locality.

We thank the reviewer for this observation, which relates to the methodology concerns raised in **R1C5**. As detailed in our response to **R1C5** and now incorporated into the revised Methods section (Lines 463-470), we performed a sensitivity analysis using three additional, finer grid resolutions (500 m, 250 m, and 100 m) to evaluate the effect of neighborhood definition on compliance rates. This analysis (Figure S7) demonstrated no significant differences in our overall findings, supporting our use of the 1 km grid.

Furthermore, we respectfully clarify that the 30% rule is specific to tree canopy coverage (“30% tree canopy in every neighborhood”). Therefore, our analysis focuses on the trees *within* parks and public spaces, rather than the total park area itself. We believe this focus maintains fidelity to the specific definition of the 3-30-300 framework. We have also clarified this strict focus on tree canopy, and the subsequent exclusion of total vegetated areas, in the Limitations section (Lines xxx) of the revised manuscript.

14. Methods - Rule 300 – line 376 – The resolution is 50 m. This part of the research is at a completely different level to the 3-trees rule (50 m versus 500/250 m).

The reviewer is correct in noting that the layer used for defining our green urban areas has a resolution of 50 meters. Our approach involved first identifying the 50-meter pixels classified as Urban Vegetation, Green Urban Areas, and Sport and Leisure Green. These areas were then converted from raster to polygons using the appropriate QGIS function. We only considered areas of at least 1 hectare in size as our urban green areas. The resulting shapefile was used for the Rule 300 analysis (Lines 479-483):

“From this map, we first extract the pixels corresponding to the Urban Area (pixels below 2000 as classification), then isolated areas classified as Urban Vegetation, Green Urban Areas, and Sport/Leisure Green within those urban areas. We then converted these areas into shapefile polygons using the polygonize function in QGIS and kept only those that were at least 1 hectare in size, defining them as our green urban areas.”

Switching to shapefiles and their 300-meter buffers resolves problems caused by different data resolutions. This doughnut-shaped area was used to calculate the number of people living within a 300-meter buffer around urban green areas, utilizing the 100-meter resolution GHS-POP data (Lines 483-486):

“We created 300-meter buffer zones around the delineated parks and green spaces to map the accessible green area for nearby residents. We then calculated the percentage of the city's population that lives within these zones to show which communities benefit from green space proximity.”

15. Lines 398-410 - This part is more of a discussion than a method.

We thank the reviewer for the suggestion, we have added some of this information in the new section on challenges and limitation (Lines 313-320):

“Finally, distances in the Rule 300 were calculated using the “as the crow flies” method, based on Euclidean measurements. While this approach is straightforward and efficient, it may not accurately account for physical barriers like buildings or roads without pedestrian crossings. To validate our method, we compared it with recent research by DG REGIO of the European Commission⁵⁸, which investigated access to urban green spaces within a 10-minute walking time along the street network. Our findings demonstrate a strong correlation between the two approaches (p -value $\ll 0.01$, $R^2 = 0.37$, see Methods and Figure S6), highlighting the robustness of our methodology.”

16. Methods - Disposable Income - The data being compared here have different resolutions. I understand that it is problematic to have the same data for different countries. However, the disadvantages of comparing data at different levels of detail must be clearly explained.

We appreciate the reviewer's concern regarding the comparison of disposable income data at varying resolutions. We acknowledge that the direct comparison of data with differing

levels of detail can introduce methodological challenges, particularly concerning potential aggregation bias or misinterpretation of scale effects.

However, our approach is not a simple comparison but a deliberate multi-scalar strategy designed to test the robustness and scale-invariance of the relationship between wealth and access to green space. We respectfully contend that this method is a strength of the study, enabling a more comprehensive understanding of the observed trends:

- Pan-European Scale (1 x 1 km resolution): We established the broad relationship using the highest resolution available data for a continental assessment (Mikou et al., 2025 and Figure S2).
- Granular Analysis (Country/City Levels): We then utilized available higher-resolution data for select countries and cities (e.g., 200 m x 200 m in France) to demonstrate that the socioeconomic disparities persist across finer scales.

This layered examination of the data is crucial for several reasons. Firstly, it enables the examination of socioeconomic patterns at a micro-level, revealing disparities that broader analyses might overlook and identifying neighborhoods with significant variations in income levels and access to green spaces. Secondly, granular analysis uncovers local trends and correlations, providing detailed insights into unique patterns within specific cities or regions that larger-scale studies might miss. Additionally, granular data minimizes aggregation bias, offering a more accurate representation of the socioeconomic landscape, which is essential for evaluating the impact of urban planning and development policies. This information has been incorporated into Lines 227-236 of the revised manuscript:

“Examining data at such a finer scale not only corroborates broader trends but also enhances our understanding at the local level, offering a more nuanced and reliable insight into the relationship between income and urban greenness. Granular data significantly reduces aggregation bias, thereby providing an enhanced representation of the socioeconomic landscape, which is crucial for evaluating the impact of urban planning and development policies. The case of Lyon serves as a compelling example of this approach, illustrating the value of detailed analysis in capturing the complexities of urban dynamics. When assessed at the national level (Figure 4b), a clear trend emerges across all three countries, reinforcing the consistent association between higher disposable income and the likelihood of residing in greener urban neighborhoods.”

By demonstrating consistent findings across these varied and nested resolutions, we mitigate the perceived disadvantages of mixed resolutions and significantly enhance the reliability and generality of our conclusions.

In summary, we believe that the main strengths of this article are its focus on the important social issue of quality of life in cities and its analysis of access to green spaces using the 3-30-300 concept, which has recently gained popularity due to its apparent simplicity. The main drawback of this article is that it uses the 3-30-300 concept for European-scale research, utilising relatively low-detail data, whereas the essence of this concept lies in its local application (e.g. proximity to green spaces and residences). Therefore, in my opinion, the results obtained, mainly regarding the three-tree visibility requirement, raise serious

doubts. Direct tree visibility is measured using parameters that are larger than the accessibility to large parks (measuring points every 500 m and visibility of 250 m, while access to the park in a distance of 300m). These are not comparable principles and should not be analysed together. However, in order to preserve the article's value, I propose rewording it to clarify the problems and inaccuracies arising from the aforementioned flaws. At the same time, we should avoid making overly strong, unequivocal statements that are not supported by the results obtained. The description should reflect the fact that the results are not entirely precise.

We appreciate the reviewers for their insightful comments and suggestions, which have been instrumental in refining our manuscript, as detailed in our point-by-point response. In particular, we appreciate the reviewer's acknowledgement of the study's strengths regarding its focus on quality of life and the crucial application of the 3-30-300 framework.

We agree that the concept is inherently local, but we respectfully contend that adapting the framework for European-scale research is a primary strength and novelty of our work, not a drawback. Our goal was to test the applicability of these principles for large-scale urban policy, which necessitates data that is consistently available across diverse national contexts.

As detailed in our response **R1C16**, our analysis is structured as a multi-scalar study. We utilize the highest available consistent resolution for the pan-European analysis (1 km grid) but validate the stability of our findings using higher-resolution data at the country and city levels (down to 200 m in some cases). This strategy demonstrates that the underlying trends are scale-invariant, thereby strengthening the reliability of the aggregated results. As outlined in our response to Comment **R1C11** (and answer to comment **R3C5**), we refined our analysis on Rule 3 by implementing a sensitivity analysis on the viewshed parameters. This enhanced the resolution of the methodology to 100 m}, which represents the highest possible spatial resolution achievable with the 100 m input data grid. By using these most refined parameters (d=100 m, r=100 m), we ensure our analysis maintains the local fidelity required for the Rule 3 while contributing to the overall, multi-dimensional assessment of urban nature as intended by the 3-30-300 framework.

As outlined in our response to Comment **R3C5**, our focus is intentionally on population units—aligning with the 3-30-300's goal of assessing social equity and human exposure. We mitigate aggregation issues by using the 100 m resolution GHSL-POP data, a method designed to provide a more refined view of residential distribution than is typical in large-scale studies.

In line with the proposal to avoid overly strong statements and ensure precision, we have carefully reviewed the discussion and results sections. We have incorporated specific language in the revised manuscript (Section **Challenges and limitations**) to reflect the inherent complexities and assumptions that arise when adapting a local framework like 3-30-300 to the continental scale.

We hope that our revisions and responses have effectively clarified our methodology, allowing us to draw robust and contextually defined results and conclusions.

Reviewer #2 (Remarks to the Author):

Reviewer #3 (Remarks to the Author):

R3C1: This is a straightforward and well-written paper, offering what is likely the largest 3-30-300 study to be published in terms of its spatial scale, offering novelty in its coverage of the entirety of Europe's urban regions. It conclusively supports the insights offered by past studies: urban residents have poor access to green space, and this is a consistent problem across very diverse urban regions. I appreciated the examination of wealth and aridity as mediating factors, this was a valuable progression on the base message. The data visualization was also a strength. I really enjoyed reading this paper and had very few concerns with the Main section; these are detailed below.

We sincerely appreciate the reviewer's positive feedback on our manuscript. We're delighted that the paper's extensive scale and novel coverage of Europe's urban regions, along with our analysis of wealth and aridity, were well-received. Thank you for recognizing the quality of our data visualization and for your thoughtful insights.

R3C2: However, on reaching the Methods I came to understand how coarse this analysis is, and also felt that some of the care in the previous sections was missing, with a few key decisions in the method lacking justification. All three 'tests' for the 3-30-300 are conducted at a very coarse spatial scale, with quite coarse data. Canopy data is at 10m resolution, neighbourhoods for canopy assessment are 1km x 1km blocks and viewsheds are based on 'observation points' at very low density per square kilometre. Tree point locations are from 2019. Park access was also calculated as a 'crow flies' distance using buffers rather than walking distance along footpaths. Some simplification of methods is understandable given the scale of the study, but my concern is that the research must remain consistent with the intent of the 3-30-300 rule, which aims to localise access to urban nature and pull focus to what is accessible from buildings. The methods applied here feel more aggregated, with no stage of the method using buildings as a direct unit of testing.

We appreciate the reviewer for highlighting these valid points regarding our methodology. These feedbacks have been invaluable in helping us articulate arguments that were previously underrepresented and in enhancing the robustness and consistency across the various methodologies and datasets used. We hope that the revised Methods section more clearly explains our reasoning and assumptions. We have addressed the issues raised in the responses to the comments below.

R3C3: The 3 test's viewshed method was difficult to follow and may benefit from some interpretive visuals.

We substantially reframe the Rule 3 section of the methods providing more details about our methodology and choices in the main text (see L. 88-99) as:

"Konijnendijk (2023)¹⁸ suggests that every resident should have a view of three trees from their dwelling, educational facility and workplace. However, defining this visibility is challenging. Various studies employ different methods: some use surveys and window-view analyses to assess tree visibility from households^{31,32}, others utilize computer vision to quantify street greenery³³, and some define tree visibility using buffer zones with distance thresholds^{19,20,34}. Here, to assess the visibility of three trees, we used viewshed

analysis^{31,35,36} (see Methods). First, we mapped areas visible from specific observation points, accounting for obstructions like building height and surrounding terrain features. We then integrated these maps with high-resolution (3 m) tree count data from PlanetScope^{27,28} (see Methods) to focus on raster cells containing at least three trees. Finally, we applied this layer of visual green exposure to the Global Human Settlement Layer Population³⁷ (GHS-POP) data to estimate the population residing in areas with visibility of at least three trees. “

Furthermore, we have also added Figure S11 to facilitate interpretation:

Figure S11: Example of results extraction for Rule 3 with a minimum distance between points of 100 meters and an analysis radius of 100 meters. Panel (A) shows the binary viewshed analysis, where black pixels indicate visible areas from the yellow viewpoints. White areas are not visible due to obstacles provided in the Built-up height layer (Panel D). Panel (B) displays the number of trees identified per pixel. In this panel, we isolated pixels with at least three trees and combined this new binary layer with Panel (A). From these two rasters, we retained only the pixels with a value of 2, indicating that they contain at least three trees and are within visible areas (Panel C). This raster is then intersected with the population distribution obtained from the Global Human Settlement Layer (GHS-POP, Panel E), to obtain the final dataset of population seeing at least three visible trees (Panel F). All the analyses are conducted on rasters with 100 m of spatial resolution.

R3C4: Why are you measuring which 100m x 100m pixels are visible within viewsheds that are >500m apart? This does not appear to be a suitable proxy for testing sightlines from buildings. It was also unclear how viewsheds were selected, and why 100m x 100m pixels were opted for. My concern is that this does not really indicate whether the buildings have views to trees. This highlights a key challenge in the method.

Our analysis was conducted at 100 m of resolution as common bases for all the three-key dataset: building height, population distribution and tree count (aggregated from 3 m resolution). We specified this in the revised section of Methods as follows (Lines 363-380):

“The analysis was carried out using the QGIS 3.30.2 version and required three different datasets: distribution of building height, estimates of tree cover count and population

distribution (GHSL-POP). Detailed information and the resolution of these layers are reported in Extended Data Table 1.

The spatial information about building height was retrieved from the Global Human Settlement Layers website (citation). The spatial raster dataset illustrates the distribution of building heights derived from global digital elevation models (DEMs) and filtered satellite imagery using linear regression. Two key DEMs were used: ALOS World 3D - 30m (from 2006-2011) and Shuttle Radar Topography Mission 30m (from 2000). The building height information was updated with shadow data from the 2018 Sentinel-2 satellite image, and the final product (GHS-BUILT-H) is provided at 100 m of resolution.

Similarly, the Global Human Settlement Layers provided the distribution of residential population, expressed as the number of people per cell. Residential population estimates for the year 2020 were disaggregated from census or administrative units to 100 m resolution grid cells, informed by the distribution, volume, and classification of built-up as mapped in the Global Human Settlement Layer (GHSL) global layer per corresponding epoch. Therefore, it can be reasonably assumed that when the value of a grid cell is greater than zero, it indicates the presence of residential buildings where people actually reside.”

To improve the robustness of our results we have conducted a sensitivity analysis of the minimum spacing between viewpoints and their respective radius of analysis (see L. 412-424 and 435-443):

“The first step of the viewshed analysis consisted in defining the number of observation points from which trees are supposed to be visible. Thus, for each city we created a set of viewpoints randomly sampled with a minimum spacing distance, namely a point will not be added if there is an already generated point within this (Euclidean) distance from the generated location. We selected three minimum distance thresholds of 100 m, 250 m, and 500 m. Lower thresholds could not be selected due to the spatial resolution of the analyzed raster layers (i.e., 100 m). The next step consisted in setting the radius of analysis, which is the maximum distance for visibility testing for the viewshed algorithm. Here we used two values 100 m and 250 m, with the observer height set to 1.6 m. Then, we run the viewshed analysis to define as visible those portions of landscape without obstructions (i.e., buildings or sloped terrain) from the building height layer, GHS-BUILT-H. Finally, visual exposure values from viewsheds generated from all sampling points are added, resulting in a continuous raster map of visual exposure”.

“The population in urban areas with visibility of at least three trees has been calculated using four distinct parameter sets, specifically exploring three values for the minimum distance between viewpoints ($d = 100\text{m}, 250\text{m}, 500\text{m}$) and two values for the radius of the viewshed analysis ($r = 100\text{m}, 250\text{m}$). The findings of this sensitivity analysis are illustrated in Figure SX2. In the main text, we chose to present the results that offer the highest viewpoint density and the most refined spatial resolution, achieved with $d = 100\text{m}$ and $r = 100\text{m}$. Further refinement of the spatial scale was constrained by the limitations in the resolution of the input datasets. With this parameter combination, 46.7% of the population meets the criteria specified by rule R3.”

Figure S6. Sensitivity Analysis of Population Visibility of Trees (Rule R3). The scatter plot illustrates the range of estimated population shares meeting the Rule R3 criteria (visibility of at least 3 trees) across 862 European cities, based on four distinct parameter sets for the viewshed analysis. The analysis varied the minimum distance between viewpoints (d : 100m, 250m, 500m) and the viewshed radius (r : 100m, 250m). The final parameter set chosen for the main text results— $d = 100\text{m}$ (highest viewpoint density) and $r = 100\text{m}$ (most refined spatial resolution)—yielded the maximum population share meeting the criteria, with an estimated 46.7% of the population having visibility of at least three trees. This selection offers the highest feasible spatial resolution, constrained by the input dataset limitations.

As detailed in the new paragraph on method limitations, we were unable to achieve a resolution finer than 100 meters due to the constraints of the data layers employed. However, this limitation could be addressed by future research and advancements in satellite datasets. While finer resolution data and local information are available for some cities, allowing for more detailed mapping of the 3-30-300 rule, our analysis focuses on consistently applying this rule across more than 800 cities on a continental European scale for the first time.

R3C5: The 3-30-300 method really looks at buildings, not population units. It seems that use of population units as the basis of the method produces significant issues of aggregation. Buildings, by contrast, can be tested individually, as has been demonstrated in past large 3-30-300 studies.

We appreciate the reviewer's insight regarding the unit of analysis. We respectfully clarify that the 3-30-300 rule is fundamentally centered on ensuring social equity and public health benefits for people. Therefore, our focus on population units aligns with the core goal of the framework: investigating how residents experience tree visibility, coverage, and proximity.

While many prior studies have focused on buildings for methodological convenience, we sought to overcome this limitation and provide a more direct assessment of human exposure to urban greenness.

To mitigate the aggregation issues associated with population data, we utilized the Global Human Settlement Layers Population (GHSL-POP) dataset, which disaggregates population estimates to a fine 100 m resolution grid informed by the built-up areas mapped in the Global Human Settlement Layer (GHSL). This detailed spatial resolution helps bridge the gap between population units and building-level analysis by providing a more granular view of residential distribution. The underlying GHSL data considers the presence and density of built-up areas and buildings within each 100 m grid cell, providing a crucial link between where people reside and the urban structures they inhabit. While GHSL-POP may not directly indicate views to trees for individual buildings, it offers a refined approach to understanding how the population is distributed in the context of urban structures.

By focusing on people, our study addresses a crucial gap, as few 3-30-300 studies have had this explicit, population-centric focus. Therefore, while GHSL-POP may not eliminate all challenges associated with the method, this robust data resolution minimizes aggregation bias and provides a valuable layer of detail that enhances the assessment of population exposure to urban green spaces. We believe this methodological choice strengthens the policy relevance and social impact assessment of our findings.

R3C6: 1km neighbourhood blocks for the 30 test are also not justified and this part of the method seems an especially severe deviation from the intent of the metric. At this scale, individual buildings could be hundreds of metres from significant patches of canopy, while still passing the '30' test. The authors are correct that 'neighbourhood' is not clearly defined – a key issue with the method itself – but this does not necessarily imply that a highly aggregated spatial scale is appropriate. I would strongly recommend a finer scale for this step of the analysis, as well as a clear justification of the scale selected. One option is to select a distance at which thermal effects of canopy – or perhaps a selected health benefit – are detectable.

We appreciate the reviewer's feedback highlighting the limitations and potential improvements in our methodology regarding the 30% tree cover rule. In response, we have revised the methods section (see Lines 463-470) to include a sensitivity analysis using three additional grid sizes as proxy for our neighbourhood definition: 100m, 200m, and 500m.

“To further assess the sensitivity of our results to the neighborhood definition as a 1 x 1 km grid, we calculated overall compliance with the 30% rule using three additional grids with resolutions of 500 meters, 250 meters, and 100 meters. Our sensitivity analysis (Figure S7) indicates that the choice of grid size does not show relevant differences in the overall compliance rates with the 30% Rule. For this reason, and in accordance with specific literature that often defines an egocentric neighborhood using a 500 m radius around each building^{18,38,39,40,63}, which we consider comparable to our 1 x 1 km approach, we decided to present the results evaluated at the 1 km grid (Figure 1b).”

We have also added a dedicated paragraph to the Limitations section (Lines 305-313) detailing the methodological considerations of the 30% rule.

“Regarding Rule 30%, it is important to note the absence of a standardized neighborhood definition; therefore, we conducted a sensitivity analysis across various neighbourhood sizes to evaluate the effect on the number of people in accordance with the 30% Rule (see Methods and Figure S7). Additionally, some studies incorporate all vegetated areas, including parks, whereas our analysis does not. Furthermore, it is crucial to acknowledge the rule's omission of blue spaces, which have been demonstrated to provide similar benefits to green spaces, including restoration, facilitating social interactions, and lowering psychological distress^{55,56,57}.”

These adjustments aim to provide a more comprehensive and reliable assessment of our findings.

Figure S7: Sensitivity analysis of the proportion of the population in cities complying with the 30% rule across different grid sizes (i.e., varying neighborhood sizes). Each point represents a city in our database, with black horizontal dashed lines indicating the average compliance with the rule. At the top, we provide an example from the city of Thessaloniki, Greece, where black pixels denote "neighborhoods" achieving 30% tree cover.

R3C7: The 300 test's use of buffers is less of a serious problem but must be better signposted in the Main text (e.g. at line 126,127 the term 'buffer' could have been inserted as a cue to the method used). A discussion of how this serves to overestimate access would also be appropriate, ideally backed by a short sample analysis demonstrating the scale of

overestimation/uncertainty this introduces to the study relative to more precise network-based routing analysis methods.

We thank the reviewer for the recommendation. First, we have edited the specific part in Lines 134-137 as:

“To examine the third rule, we utilized spatial buffering to estimate the proportion of the population residing within a 300-meter zone surrounding parks and other urban green areas larger than 1 ha (see Methods). This buffer serves as an indicator of the accessible green area for nearby residents.”

Furthermore, in the new section on limitations and challenges we report more information on the methodology used for Rule 300, and we discuss the limit of our approach and we also show a comparison with a more refined criteria for defining distance to urban green areas (see Lines 309-320 and Lines 495-508 in Methods):

“Additionally, some studies incorporate all vegetated areas, including parks, whereas our analysis does not. Furthermore, it is crucial to acknowledge the rule's omission of blue spaces, which have been demonstrated to provide similar benefits to green spaces, including restoration, facilitating social interactions, and lowering psychological distress^{55,56,57}. Finally, distances in the Rule 300 were calculated using the "as the crow flies" method, based on Euclidean measurements. While this approach is straightforward and efficient, it may not accurately account for physical barriers like buildings or roads without pedestrian crossings. To validate our method, we compared it with recent research by DG REGIO of the European Commission⁵⁸, which investigated access to urban green spaces within a 10-minute walking time along the street network. Our findings demonstrate a strong correlation between the two approaches (p -value $\ll 0.01$, $R^2 = 0.37$, see Methods and Figure S8), highlighting the robustness of our methodology.”

Comparison of Population Access to Green Areas

Comparing two databases based on the share of population

Figure S8: Comparison between the share of European cities' population meeting the R-300 rule with the proportion of urban center populations having access to at least 1 hectare of green urban areas within a 400-meter walk, as reported in the DG REGIO study.

R3C8: Overall I am quite concerned that the use of population has broken the method. Ideally this study would carry out building-scale tests, then link these results to population by making a statistical argument around how many people each building might represent. This avoids the need to use very aggregated methods which introduce many of the problems that 3-30-300 was conceived to avoid.

We thank the reviewer for the recommendation, we have address similar issue in our answer to comment **R3C5** above.

Minor issues in the Main text:

R3C9: Line 76 – What's LUISA?

We spelled out the acronym LUISA in the main text as “Land Use-based Integrated Sustainability Assessment”. In particular, the LUISA Base Map 2018 is a high-resolution land use/land cover map developed and produced by the Joint Research Centre of the European Commission (see Citation 25). It corresponds to a modified and improved version of the

CORINE Land Cover 2018 map. Compared to CORINE the LUISA Base Map delivers a higher overall spatial detail and finer thematic breakdown of artificial land use/cover categories (17 categories instead of 11 in CORINE).

R3C10: 85 – explanation of views is unclear.

We apologize for any confusion caused by the initial wording. We have revised the first paragraph (Lines 88-99) of the section titled "Half of the European urban population views 3 trees" to enhance clarity:

“Konijnendijk (2023)¹⁸ suggests that every resident should have a view of three trees from their dwelling, educational facility and workplace. However, defining this visibility is challenging. Various studies employ different methods: some use surveys and window-view analyses to assess tree visibility from households^{31,32}, others utilize computer vision to quantify street greenery³³, and some define tree visibility using buffer zones with distance thresholds^{19,20,34}. Here, to assess the visibility of three trees, we used viewshed analysis^{31,35,36} (see Methods). First, we mapped areas visible from specific observation points, accounting for obstructions like building height and surrounding terrain features. We then integrated these maps with high-resolution (3 m) tree count data from PlanetScope^{27,28} (see Methods) to focus on raster cells containing at least three trees. Finally, we applied this layer of visual green exposure to the Global Human Settlement Layer Population³⁷ (GHS-POP) data to estimate the population residing in areas with visibility of at least three trees.”

R3C11: Figure 3 – this is nice but a little confusing. Is the x-axis ‘share of population’... the share of the population passing the 3-30-300 test? Please annotate this axis fully. This confusion is partly sparked by line 190-191 which state the boxes show the IQR of income values. I also suggest you change the aridity quartiles to abcd instead of 1234 so that it’s easy to discern what ‘2-4’ means (at the moment it’s tricky because one is not sure whether aridity is first or second and this takes a minute to confirm).

Thank you for the suggestions and for noticing the mistake in the caption for description of the IQR values. Yes, in the x-axis we are reporting the share of the population fulfilling the 3-30-300 rule. We incorporated the edits in the updated version of Figure 3.

Figure 3: Relationship between economic development, climate, and urban greenness in 862 European cities. (a) Spatial bivariate representation of per capita GDP (GDPc) and aridity index, where circle size represents total city population and discrete color indicates the 4th quantile of each distribution. (b) Boxplot analysis of the 16 resulting groups, showing the correlation between GDPc, aridity index, and the percentage of the city's population fulfilling the 3-30-300 rule. The horizontal dashed lines separate the four quartiles of GDPc, while the sub-groups reveal the impact of increasing humidity on urban greenness. Notably, cities with higher GDPc tend to be located in areas with low aridity and a higher proportion of their population living in greener areas, suggesting a positive relationship between economic development and urban environmental quality. Each box depicts the interquartile range (IQR) of the percentage of the city's population fulfilling the 3-30-300 rule with the median marked by a line inside the box. The whiskers extend to the most extreme data points within 1.5 times the IQR, highlighting the spread of the data. Outliers beyond the whiskers, if any, are plotted individually.

R3C12: Figure 3 also does appear to show that aridity is a major driver of low 3-30-300 achievement – the box for 3-2 seems to indicate that in the third income quartile, the lowest humidity cities have much lower achievement. The text at lines 177-180 appears to deny this.

The reviewer rightly highlights that aridity significantly contributes to low achievement levels. Across all four income classes, it's evident that more arid cities tend to have a lower percentage of their population meeting the 3-30-300 rule. We have revised the relevant section to emphasize and clarify this point (see L. 191-195):

“Another crucial factor influencing this relationship are humidity-aridity conditions. Cities located in more humid environments tend to have a greater share of green spaces, which results in a higher adherence to the 3-30-300 rule. Notably, we showed that on average,

within the same GDP per capita classes, cities with higher humidity levels provide residents with more access to green spaces (Figure 3b)."

R3C13: 203-210. Please justify why a more granular analysis was necessary.

Thank you for your feedback, which has allowed us to clarify the importance of enhanced granularity in our analysis. This approach is crucial for several reasons. Firstly, it enables the examination of socioeconomic patterns at a micro-level, revealing disparities that broader analyses might overlook and identifying neighborhoods with significant variations in income levels and access to green spaces. Secondly, granular analysis uncovers local trends and correlations, providing detailed insights into unique patterns within specific cities or regions that larger-scale studies might miss. Additionally, granular data minimizes aggregation bias, offering a more accurate representation of the socioeconomic landscape, which is essential for evaluating the impact of urban planning and development policies. This information has been incorporated into Lines 227-236 of the revised manuscript:

"Examining data at such a finer scale not only corroborates broader trends but also enhances our understanding at the local level, offering a more nuanced and reliable insight into the relationship between income and urban greenness. Granular data significantly reduces aggregation bias, thereby providing an enhanced representation of the socioeconomic landscape, which is crucial for evaluating the impact of urban planning and development policies. The case of Lyon serves as a compelling example of this approach, illustrating the value of detailed analysis in capturing the complexities of urban dynamics. When assessed at the national level (Figure 4b), a clear trend emerges across all three countries, reinforcing the consistent association between higher disposable income and the likelihood of residing in greener urban neighborhoods."

By demonstrating consistent findings across these varied and nested resolutions, we mitigate the perceived disadvantages of mixed resolutions and significantly enhance the reliability and generality of our conclusions.

R3C14: 231 – I object to the use of 'inevitably'. Surely there are areas where green access is good even in areas which have urbanised. This undermines a core message that is important in this field of research: urbanisation does not need to mean huge losses of greenery. The problem is one of design; we have agency in this. The term inevitably implies that infill development is an unquestionable force of 'greying'. To date it mostly has been, and perhaps this is a better way to acknowledge the issue, without creating a sense of helplessness.

Thank you for the suggestion, we rephrased the sentence in Lines 252-257 as: "While urbanization is often seen as a positive force driving economic growth and resource efficiency, it can impact the availability and quality of urban green spaces if not managed thoughtfully. Addressing this challenge through intentional design and planning is crucial to creating sustainable and livable cities"

R3C15: 241-257. These are good arguments but streets are a minority of urban space, often 30-40%. What other land uses can play a role in greening? Streets alone won't get us to 30.

We thank the reviewer for the suggestion. We have revised the section to offer more insights on how cities can overcome these challenges to achieve the 30% cover target. Our recommendations include prioritizing peri-urban forests, expanding tree planting initiatives to include private spaces, aiming for 30% overall vegetation with a substantial tree component in densely built and arid areas, and creating accessible pocket-sized parks in highly urbanized settings. We specify this in Lines 283-298 as:

“In this context, Lungman et al., (2023)⁵² showed that increasing tree coverage to 30% would cool cities by a mean of 0.4°C, which could prevent up to 1.84% of summer premature deaths. We acknowledge that achieving the target of 30% tree cover in densely built-up and populated cities presents significant challenges as streetscapes alone are often insufficient. To overcome this, greening strategies must utilize other major land uses. Peri-urban forests should be prioritized for their positive impact on surface climate conditions and air quality⁵³. Tree planting programs must also be expanded to private land and residential areas⁵⁴ which collectively represent a vast, often underutilized surface area for canopy growth. Greening of buildings (e.g., green roofs, green walls, and facade planting) should be a primary focus particularly in dense urban areas where ground-level space is scarce, to add crucial vertical and horizontal canopy surfaces. Although trees offer numerous benefits, achieving 30% cover solely with trees in densely built environments may be difficult, especially in arid climates. In such scenarios, implementing more pocket-sized parks with a considerable tree component can be beneficial²². These parks, created from residual land, are accessible and appealing to local communities and easier to establish.”

R3C16: 265-269. As with my comment regarding inevitability above: the argument that 30% in urban areas is unfeasible is not supported by your evidence. 30% cover is more difficult to deliver in urbanised areas but ‘spatial and environmental limitations’ are the product of design decisions; if an urban area has no space for trees then it has been designed badly. These are not feasibility problems. Indeed, the street retrofit ideas you indicate in 241-257 are promising remedies to these more dense areas, especially as these more walkable dense areas do not need to cater to drivers as generously.

We thank the reviewer for the recommendation, we have address similar issue in our answer to comment **R3C15** above.

R3C17: Finally – your title is very punchy, but it uses the term ‘green space’. I think many readers will interpret this as ‘parks’, which was my initial impression too. A less ambiguous term may help avoid this misconstrual of the study’s focus.

We thank the reviewer for the recommendation, we have changed our title in:

“Less than fifteen percent of European city residents meet the 3-30-300 rule for green space visibility, canopy, and proximity”

to better highlight the focus on the 3-30-300 rule and avoid misunderstanding.

Response to reviewers for the manuscript

Assessing European Cities with the 3-30-300 rule underscores the need for enhanced urban greening efforts

L.E. Bertassello ^{1*}, M. van der Velde ¹, J. Maes ², S. Liu ³, M. Brandt ³, L. Feyen ¹

1. European Commission, Joint Research Centre (JRC), Ispra, Italy
2. European Commission, Directorate-General for Environment (DG-ENV), Brussels, Belgium
3. Department of Geosciences and Natural Resource Management, University of Copenhagen, Copenhagen, Denmark

* Corresponding Author: Leonardo Enrico Bertassello, leonardo.bertassello@ec.europa.eu,
ORCID: <https://orcid.org/0000-0001-5168-2142>

Note: the answer to reviewers' comments are highlighted in red, and the references to the text refers to the tracked changes version of the manuscript.

Reviewer #1 (Remarks to the Author):

R1C1: Thank you to the authors for addressing the proposed revisions and carefully responding to the comments raised. The changes introduced in many places have clearly improved the clarity of the research and its presentation. Numerous parameters have been explained, formulations have been refined, the argumentation has been better structured and key aspects of the methodology have been clarified. Several previously ambiguous sections have been significantly clarified. Consequently, I now have a much better understanding of the research, and the article is more coherent and easier to follow.

We sincerely appreciate the reviewer's positive feedback on our revised manuscript.

While the explanations and arguments provided by the authors are sufficient for most of the comments I raised, two issues still require further refinement in my opinion.

R1C2: The first issue concerns Rule 1 (the visibility of three trees). The additional explanations provided by the authors greatly enhance the reader's understanding of the research process, which appears logical and interesting. However, I remain concerned about the level of detail in the analysis and the conclusions drawn from it. For instance, given the tree centroids and observation points, I do not understand why the authors first generalised the tree information by counting trees within a 100 m grid cell and only then determined their visibility. Performing this operation prior to generalising the tree data would have yielded more precise results.

We thank the reviewer for these critical observations regarding the spatial resolution of our analysis. We apologize for the confusion but the decision to generalize tree data into 100m cells prior to the visibility analysis was a deliberate methodological choice necessitated by the resolution of our auxiliary datasets.

While tree centroids were available at a higher resolution, the '**obstacle layer**' (building heights from GHS-BUILT-H) and the **population distribution data** (GHS-POP) are only available at a 100m grid resolution (see updated Lines 314-316). In a viewshed analysis, the precision of the output is constrained by the coarsest input layer, and to the best of our knowledge, at pan-European scale, the most reliable building height data (GHS-BUILT-H) is currently restricted to a 100m resolution. We added this limitation in L. 459-470 of the method section as:

“Further refinement of the spatial scale was limited by the resolution of the input datasets. In the viewshed analysis, the precision of the output is defined by the coarsest input layer, and to the best of our knowledge, at pan-European scale the most reliable building height data (GHS-BUILT-H) is currently restricted to a 100m resolution. Therefore, using data on tree crown centroids and observation points distributed more densely (e.g. 10m) against a 100m obstruction block would not yield more accurate results; rather, it would imply a level of precision that the building height data cannot support. In particular, even if we use a set of viewpoints with higher density (e.g. 10 m of distance) we would still be limited by the radius of analysis, whose extent could be at minimum the size of the obstacle layer, and therefore 100 m.”

R1C3: Moreover, I do not understand how grid cells containing obstacles that block the visibility of trees were taken into account within the adopted workflow. In particular, in the case of 100 m axis, in particular, the cells obstructing visibility may be the same as those containing the three trees. What, then, is the outcome of the analysis in such cases? What happens when the cell that blocks visibility is the same cell where the observation point falls?

Our approach considers the entire 100m cell as a single unit of analysis. If an observation point exists within a cell that also contains trees, those trees are deemed "visible" based on proximity, unless the line of sight is blocked by an adjacent obstacle cell. This approach is a simplification of the "window-view" concept, but the high density of observation points ensures visibility is assessed from multiple angles, thereby reducing the likelihood of "dead zones" where trees might not be counted.

In cases where a cell obstructs visibility but is still visible from other observation points, it will be marked as "visible" (illustrated as the black region in Figure S11(a)). Subsequently, we assess whether this visible cell contains three trees (as shown in Figure S11(b)). If it does, it will be classified as "visible with three trees" (as depicted in Figure S11(c)); otherwise, it will not be included in the final count. This methodology ensures that even when obstacles and observation points coincide, visibility can be reliably assessed across the urban landscape.

R1C4: In my opinion the results obtained in this rule are highly generalised (and thus unreliable) for another reason. The spacing of observation points at 100-metre intervals is too coarse; within a radius of 100 metres, several buildings and other obstacles may be present, leading to very different local visibility conditions. Using such large distances to assess tree visibility moves the analysis away from the core principle of the 3-30-300 concept: that greenery should be in residents' immediate surroundings. Using data on tree crown centroids and observation points distributed more densely (e.g. 10m) would make it possible to obtain fairly precise results that capture local conditions in a very straightforward

manner, which is the essence of Konijnendijk's concept. Moreover, these assumptions strongly influence the general results of the three-tree criterion assessment, yet they are addressed with only a single sentence of commentary in the limitations section.

We acknowledge the reviewer's concern regarding the coarse resolution, which indeed poses a significant limitation to the output of the three-tree criterion. Although our tree count dataset allows for analysis at a finer 3-meter resolution concerning tree visibility, we are hindered by the coarse resolution of the "obstacle layer" (specifically, the building height map, GHS-BUILT-H) and the population distribution (GHS-POP), both of which are available only at a 100-meter resolution. Enhancements in these datasets are essential for conducting a more precise analysis. While local studies have achieved greater precision with more detailed data for individual cities (Nieuwenhuijsen et al., 2022; Coreser et al., 2024; Antonio Lopez et al., 2025; Battisti et al., 2024), a comprehensive obstacle dataset at the European scale is currently unavailable. Finally, to evaluate the impact of our 100-meter resolution on the reliability of the results, we benchmarked our findings against several recent high-resolution case studies (see Table below reported also in Supporting Information as Table S2). These local studies utilized high-precision datasets and building-level vantage points. While our continental-scale approach necessarily generalizes local conditions, the resulting estimates for Rule 1 remain broadly consistent with these high-resolution assessments. The observed variations are largely attributable to two factors. First, our study measures the **share of the population** fulfilling the rule, whereas most local studies calculate the **share of buildings**. Second, while a 100-meter grain may average out specific micro-scale visibility "gaps," it effectively captures the neighborhood-level green character that underpins the 3-30-300 concept. The alignment between our pan-European results and these localized studies suggests that the 100-meter resolution serves as a robust proxy for identifying broad spatial trends across the continent.

We have included this observation in line 312-325 of the limitations section as:

"A primary limitation of our assessment of Rule 1 (the 'three-tree' criterion) is the 100m spatial grain of the GHS-BUILT-H and GHS-POP datasets. While the 3-30-300 rule is ideally evaluated from the micro-scale perspective of residential windows, the current lack of uniform, high-resolution pan-European datasets for building geometry and population distribution necessitates a coarser approach. This 100m generalization may not capture micro-scale visibility nuances, such as individual trees visible through narrow gaps between buildings. To evaluate the impact of this generalization, we benchmarked our results against several high-resolution local studies^{19,20,21,22}. Despite the differences in spatial grain, our findings are broadly consistent with these localized assessments. This alignment suggests that while our results represent a more generalized scale, they effectively capture the overarching spatial trends of urban forest visibility. This trade-off is essential for maintaining methodological consistency across the European continent; while local studies offer superior precision for individual cities, our approach provides the first harmonized, comparable baseline at a continental scale."

City	Reference	Local Study (High-Res)	This Study (100m)	Methodological Difference
Barcelona (ES)	Nieuwenhuijsen et al. (2022)	43%	31%	Local study used individual surveys/perception.
Amsterdam (NL)	Coreser et al. (2024)	50%	46%	Local study: % of Buildings; Our study: % of Population.
Aix-en-Provence (FR)	Antonio Lopez et al. (2025)	68%	62%	Local study: % of Buildings; Our study: % of Population.
Florence (IT)	Antonio Lopez et al. (2025)	38%	59%	Local study: % of Buildings; Our study: % of Population.
Turin (IT)	Battisti et al. (2024)	40%	48%	Local study: Neighborhood-level average.

R1C5: The second issue relates to the language used to describe the conclusions. I raised this point in my first review. Many of the formulations, especially given the high level of generality of the analyses, are overstated. In other words, the conclusions drawn are too strong and radical given the level of detail in the data used and the results obtained. The authors use overly strong wording which is often not fully justified. Such statements might be appropriate if the level of detail in the data corresponded to the assumptions of the concept. However, since this is not the case, the research results require a more critical and cautious interpretation.

E.g.:

“Less than 15% of the population in European cities...”

“A quarter of city-dwellers in Europe...”

“Half of the European urban population...”

If the study only analyses 800+ cities in Europe, whereas there are around 2,000 cities in Germany alone, then it still only covers a portion of the population in European cities. While

it does represent the populations of the largest and most urbanised cities, it does not represent the entirety of the urban population.

We thank the reviewers for the comments. We have revised some parts of the manuscript where we believe the tone was overly strong. For example:

Title: Assessing European Cities with the 3-30-300 rule underscores the need for enhanced urban greening efforts

Subtitles in the Results and Discussion sections:

L 89: Nearly half of residents in the studied cities meet the three-tree visibility criterion

L 117: One-quarter of the analyzed population live in areas with 30% tree cover

L 138: Over half of residents in the analyzed cities meet the 300-meter proximity target

L 152: Less than 15% of the population in the analyzed cities fulfills the 3-30-300 rule

We further edited other sentences in the main text as in L. 19, 24, 102, 124, 143, 156, 166, 169, 270.

R1C6: In summary, while acknowledging the amount of work involved, I recommend recalculating the three-tree criterion using observation points that are more densely distributed. This would substantially increase the value of the article and ensure that the conclusions drawn are well-founded.

We thank the reviewers for the comments and suggestions that have helped make our work clearer and more well-founded. We hope we clarified the issue regarding the three-tree criterion and the reason why with the current dataset limitation (building height and population layer at 100 m resolution) we cannot enhance the granularity of our analysis as 100 m represent the highest resolution of the obstacle layer which is necessary to first run the viewshed algorithm and then compute the population living within this visible area.

Reviewer #2 (Remarks to the Author):

Reviewer #3 (Remarks to the Author):

Thanks for appropriately acknowledging the limitations of this study, and producing the sensitivity analyses that show relatively low median differences. I feel this is now ready for publication, and will likely cite it shortly. Thanks for an important piece of work.

We are grateful to the reviewer for their encouraging final remarks and for recognizing the importance of this study. We appreciate the thorough review process, which significantly improved the clarity and foundation of our work